# Multiple inputs ensure yeast cell size homeostasis during cell cycle progression

**Cecilia Garmendia-Torres**[1,2,3,4], **Olivier Tassy**[1,2,3,4], **Audrey Matifas**[1,2,3,4], **Nacho Molina**[1,2,3,4], **Gilles Charvin**[1,2,3,4]*

[1]Institut de Génétique et de Biologie Moléculaire et Cellulaire, Illkirch, France; [2]Centre National de la Recherche Scientifique, UMR7104, Illkirch, France; [3]Institut National de la Santé et de la Recherche Médicale, U1258, Illkirch, France; [4]Université de Strasbourg, Illkirch, France

**Abstract** Coordination of cell growth with division is essential for proper cell function. In budding yeast, although some molecular mechanisms responsible for cell size control during G1 have been elucidated, the mechanism by which cell size homeostasis is established remains to be discovered. Here, we developed a new technique based on quantification of histone levels to monitor cell cycle progression in individual cells with unprecedented accuracy. Our analysis establishes the existence of a mechanism controlling bud size in G2/M that prevents premature onset of anaphase, and controls the overall size variability. While most G1 mutants do not display impaired size homeostasis, mutants in which cyclin B-Cdk regulation is altered display large size variability. Our study thus demonstrates that size homeostasis is not controlled by a G1-specific mechanism alone but is likely to be an emergent property resulting from the integration of several mechanisms that coordinate cell and bud growth with division.
DOI: https://doi.org/10.7554/eLife.34025.001

*For correspondence:
charvin@igbmc.fr

**Competing interests:** The authors declare that no competing interests exist.

## Introduction

To ensure cell size homeostasis, cells must coordinate growth and division during the mitotic cycle. During the 1970s, genetic studies aimed at deciphering the biochemical architecture of the cell cycle emerged concomitantly with efforts to characterize cell size control in fission (*Fantes, 1977*; *Fantes and Nurse, 1977*) and budding yeast (*Hartwell and Unger, 1977*; *Johnston et al., 1977*). In a free-running cell cycle oscillator model (i.e. in the absence of any coupling to control signals, such as cell size), the cell division time is set by the sum of fixed intervals associated with successive cell cycle events (referred to as 'Timer'). In this case, the absence of coordination between the cell cycle engine and cell growth may induce deleterious fluctuations in cell size. In contrast, a 'Sizer' mechanism has been shown to operate in yeast: the transition to a given cell cycle phase (resp. mitotic entry in fission yeast and DNA replication in budding yeast) occurs when cells have attained a critical size during the preceding phase (resp. G2 phase in fission yeast and G1 in budding yeast) (*Fantes, 1977*; *Fantes and Nurse, 1977*). In this case, small cells experience a size-dependent cell cycle delay and therefore a compensatory mass addition works as a counteracting force to restore size equilibrium.

In the last 10 years, several important advances have been made in unraveling the molecular mechanism(s) responsible for this 'size checkpoint,' which transmits cell size information to the cell cycle control machinery. In fission yeast, it was proposed that the polarity protein kinase Pom1 localizes to the cell tips and indirectly inhibits the cyclin-dependent kinase Cdk1, allowing the cell to sense its elongation and therefore control mitotic entry (*Martin and Berthelot-Grosjean, 2009*; *Moseley et al., 2009*). Later studies disproved this model, by showing in particular that Pom1 deletion does not alter size homeostasis, as would be expected following disruption of a core player in

the size signaling pathway (*Bhatia et al., 2014*; *Wood and Nurse, 2013*). Further work proposed that Cdr2, a target of Pom1, is responsible for the coupling between cell geometry and cell cycle progression (*Pan et al., 2014*). However, this hypothesis has not yet been validated by measuring the size-compensation mechanism in the corresponding mutant background (i.e., *cdr2Δ*), and alternative models coupling cell size to mitosis have been proposed since(*Keifenheim et al., 2017*).

Early models of cell size regulation in G1 in budding yeast proposed that the commitment (called 'Start') to an irreversible round of division in response to cell growth is controlled by the cyclin Cln3, which is a key regulator of G1 progression and the function of which might be coupled to cell size in various ways (*Aldea et al., 2007*; *Wang et al., 2009*). Alternatively, recent work showed that the concentration of Whi5, a major inhibitor of G1/S cyclin expression, is gradually diluted during G1 but is synthesized in a cell size-independent manner during S/G2/M phases; thus, the nuclear concentration of Whi5 is larger in small daughter cells compared with the large mother cells at birth (*Schmoller et al., 2015*). According to this model, coupling between cell growth and cell cycle progression in G1 originates from cell size-dependent dilution of this G1/S inhibitor. Nevertheless, although the *WHI5* mutant displays a small cell size phenotype (*Jorgensen et al., 2002*), the G1 size-compensation effect is reduced but not abolished (*Soifer et al., 2016*; *Turner et al., 2012*), and the overall width of the cell size distribution of Whi5 mutants and wild-type (WT) yeast are similar (*Jorgensen et al., 2002*). Therefore, the contribution of Whi5 to the overall size homeostasis in budding yeast therefore remains a matter of debate.

*whi5Δ* mutants and cells carrying other genetic perturbations that induce a premature G1/S transition also display compensatory growth in S/G2/M (*Charvin et al., 2009*; *Harvey and Kellogg, 2003*; *Soifer and Barkai, 2014*), which is analogous to the 'cryptic' G1/S size control observed long ago in *wee1Δ* fission yeast (*Fantes, 1981*). These observations suggest that, unlike other cell cycle checkpoints (e.g., spindle assembly checkpoint) in which a single sense-and-signal machinery controls cell cycle progression, cell size homeostasis may be maintained by multiple mechanisms that cooperate to coordinate cell growth and division throughout the entire cell cycle. Adding further complexity, previous work has shown that the magnitude of the size-compensation effects during G1 is greatly affected by mutation of several genes with no direct link to G1/S signalling (*Soifer and Barkai, 2014*). This indicates that size control may result from a complex interplay between the regulatory mechanisms involved in cell cycle progression.

Recent observations in bacteria proposed that a size-compensation mechanism may not even be necessary to ensure cell homeostasis. In contrast to a Sizer mechanism, in which cell size variation during the cell cycle is negatively correlated with the initial cell size, bacteria passively reach size homeostasis through an 'Adder' mechanism, whereby a constant amount of cellular material is added at every cell cycle (*Campos et al., 2014*; *Jun and Taheri-Araghi, 2015*; *Taheri-Araghi et al., 2015*). However, as recently analyzed in budding yeast, despite the existence of a clear 'Sizer' in G1, the effective size control mechanism during the whole cell cycle may be perceived as an 'Adder'(*Jun and Taheri-Araghi, 2015*; *Soifer et al., 2016*), further raising the question of the integration of multiple size regulation steps during cell cycle progression (*Chandler-Brown et al., 2017*).

By restricting the focus to the G1 size control mechanism, most previous studies overlooked the existence of other size control mechanisms at other cell cycle stages, and, *a fortiori*, how they are integrated to ensure the overall size homeostasis throughout the cell cycle. This is in part because, unlike G1 and mitosis, others phases of the cell cycle could not be accurately resolved in single cell measurements. Therefore, a global quantitative analysis of size compensation effects during the entire cell cycle is required to determine how each phase contributes to cell size control, and how this is perturbed in mutants of cell cycle regulation. Furthermore, the strength of size control was usually assessed by simply measuring the magnitude of size compensation effects, but ignoring how the actual cell size variability – which is the key marker of size homeostasis - evolves during the cell cycle. Last, mutants in which the overall size homeostasis – and not only G1 compensatory growth - is truly impaired remain to be identified, which is decisive to improve our understanding of the genetic determinism of size control.

To address these deficits, we have developed a new microscopy technique based on real-time measurements of histone levels to monitor the successive phases of the cell cycle in individual cells in an automated manner. This methodology allowed us to measure a large number of cell cycle phase- and cell size-associated variables in 22 mutants, totaling up to 15,000 cell cycles per

genotype. Using this dataset, we quantitatively established the existence of a compensatory growth mechanism operating on the bud size during G2 in WT cells, thus confirming the existence of multiple size-dependent inputs in size control (*Harvey and Kellogg, 2003*; *McMillan et al., 1998*), in agreement with theoretical predictions (*Spiesser et al., 2015*), and clearly ruling out the 'cryptic' nature of size control in G2. Unexpectedly, among the cell cycle genes tested that affect size compensation in either G1 or G2, we found that genes related to the regulation of cyclin B-Cdk activity had the strongest impact on size homeostasis. This finding contrasted with mutants of G1 regulators, which displayed only modest effects on size control. Quantification of cell size variability during the cell cycle showed that phase-specific compensatory growth directly controls the noise strength in cell size distribution, as demonstrated using a linear map model that accommodates experimental data presented in this study. Therefore, unlike the prevailing model of a dominant G1-specific size control checkpoint, our analysis reveals that cell size homeostasis results from the integration of at least two interdependent elements acting at different stages of the cell cycle on different cellular compartments.

## Results

### A new technique to monitor cell cycle progression in live yeast cells

To obtain a precise assessment of cell size control during cell cycle progression, we sought a quantitative marker of the successive cell cycle phases in individual growing cells. Studies to date have relied on monitoring of bud emergence or of a fluorescent budneck marker, neither of which can distinguish between S, G2, and M phases. We reasoned that the burst of histone synthesis could serve as an accurate marker of S phase, thanks to the tight reciprocal coupling of DNA replication and histone synthesis, which has been characterized in detail (*Baumbach et al., 1987*; *Heintz et al., 1983*; *Nelson et al., 2002*; *Sittman et al., 1983*). Therefore, determining the onset and the end of the burst of histone expression would allow us to extract the duration of S-phase, but also deduce the duration of phases that precede (G1) and succeed DNA synthesis until anaphase onset. In budding yeast, metaphase is known to directly follow the end of replication, with no evidence of gap phase in between. However, we referred to this post-DNA synthesis interval as G2/M in the following for sake of simplicity and coherence with other organisms.

To this end, we took a strain carrying a fluorescent protein cassette fused to one of the histone 2B loci (*HTB2, Figure 1A*), which has been extensively used as a nuclear marker. We used a super-folder GFP (sfGFP) protein to ensure a fast maturation of the fluorophore, in order to prevent artifacts in measurements of histone dynamics, as described below. We monitored yeast cell growth in a microfluidic device which allowed us to track the successive divisions of individual cells forming bi-dimensional microcolonies (See Materials and methods and *Figure 1* and *Figure 1—figure supplement 1*), as previously described (*Goulev et al., 2017*).

Plotting total nuclear *HTB2*-sfGFP fluorescence as a function of time over one cell cycle (*Figure 1B*, see *Figure 1—figure supplement 2A–D* and Supporting Information for details of all quantifications), revealed a fluorescence plateau during the unbudded period of the cell cycle, followed by a linear ramp starting shortly before budding, and a plateau during the budded period of the cell cycle. This pattern was terminated by a sudden drop in fluorescence, corresponding to the onset of anaphase and nuclear division.

Quantification of fluorescence levels gave a consistent ~2-fold enrichment in histones before compared with after the histone synthesis phase (*Figure 1—figure supplement 2E and F*), as expected by the doubling of DNA content. Similarly, we verified that HTB2-sfGFP fluorescence was evenly partitioned between mother and daughter cells upon nuclear division (Daughter/Mother [D/M] asymmetry = 0.94 ± 0.01 *Figure 1—figure supplement 2*).

To further check that the measurement of total histone content over time is a reliable and physiological way to score cell cycle progression in individual cells, we performed a series of control experiments. First, we compared the division time (by measuring the anaphase to anaphase interval) of a strain carrying a constitutive NLS-GFP marker with a *HTB2*-GFP strain. We observed that the GFP-tag at the *HTB2* locus only modestly affected cell division (*Figure 1—figure supplement 3A–C*). Of note, unlike the piecewise expression pattern observed with the *HTB2*-GFP strain (*Figure 1—figure*

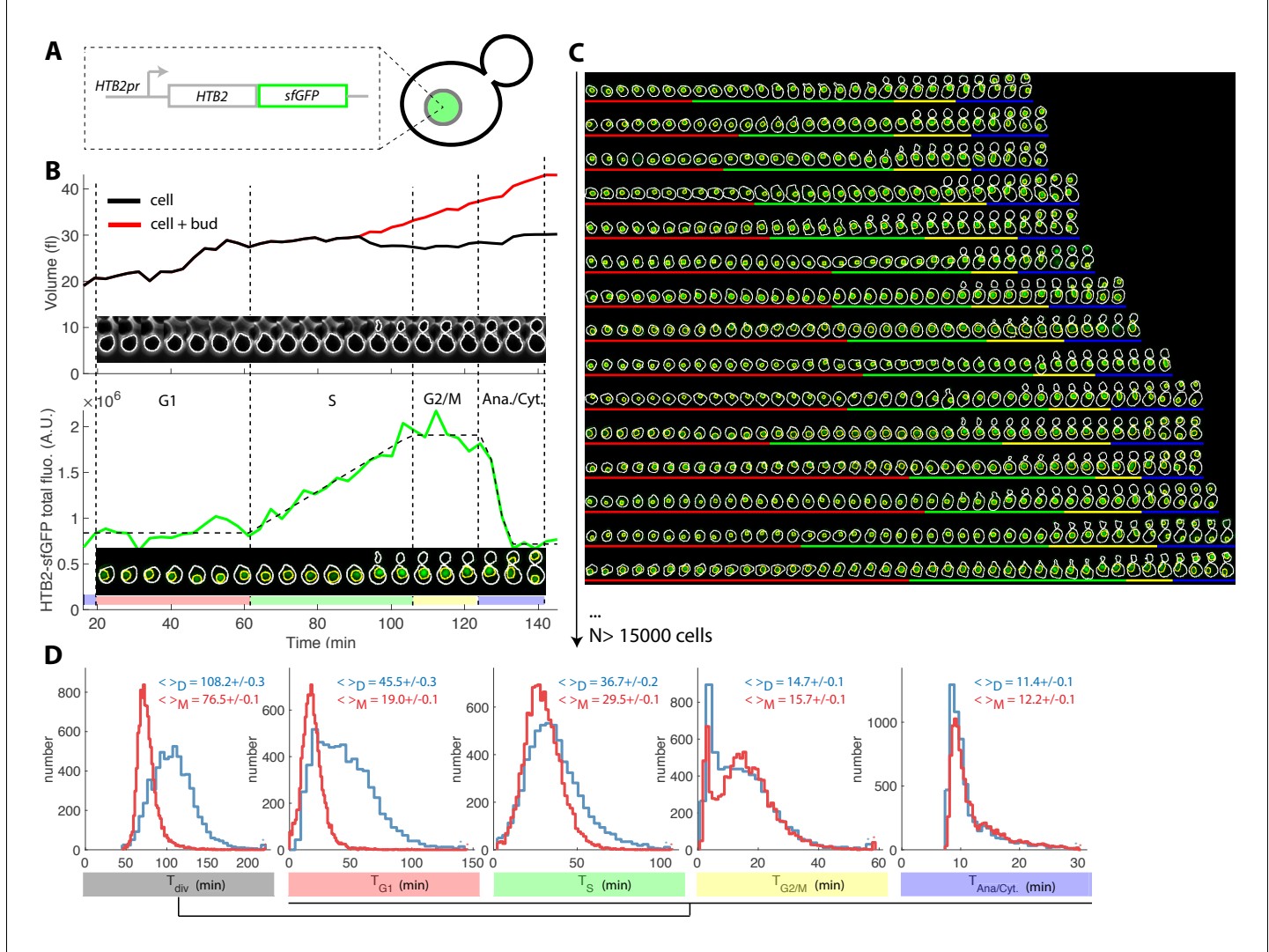

**Figure 1.** Tracking of cell cycle phases in individual cells. (A) Principle of the H2B-GFP fluorescence marker used track cell cycle progression (B) Sequence of phase contrast (upper) and fluorescence (lower) images of a sample wild-type daughter cell carrying a histone marker (HTB2-sfGFP), displayed with a 6 min interval. Segmented cell and nuclear contours are indicated in white and yellow, respectively. The upper and bottom panels show the quantification of cell (and bud) volume and total fluorescence signal (green curve) over time, respectively. The dashed line shows the best fit of a piecewise linear model to the fluorescence signal, which is used to segment the cell cycle into distinct phases (see text for details), as indicated using a specific color code. Vertical dashed lines highlight cell cycle phase boundaries. (C) Sample dynamics of 15 individual daughter cells during one cell cycle. The green signal represents nuclear fluorescence of the HTB2-sfGFP marker. White and yellow lines indicate cellular and nuclear contours, respectively. Colored segments (G1, red; S, green; G2/M, yellow; anaphase/cytokinesis, blue) indicate cell cycle intervals, as determined using the procedure described in (B); (D) Histogram of durations of cell cycle intervals and overall cell cycle for WT daughter (D; N = 6079) and mother (M; N = 10775) cells. The legend indicates the mean ±standard error on mean.

DOI: https://doi.org/10.7554/eLife.34025.002

The following video and figure supplements are available for figure 1:

**Figure supplement 1.** Principle of the microfluidic device and time lapse experiment.

DOI: https://doi.org/10.7554/eLife.34025.003

**Figure supplement 2.** Quantification of H2B-sfGFP fluorescence signal in individual cells.

DOI: https://doi.org/10.7554/eLife.34025.004

**Figure supplement 3.** Influence of the HTB2-sfGFP marker on cell cycle duration.

DOI: https://doi.org/10.7554/eLife.34025.005

**Figure supplement 4.** Effect of fluorophore maturation on the apparent dynamics of histone synthesis.

DOI: https://doi.org/10.7554/eLife.34025.006

**Figure supplement 5.** Comparison of HTB2-sfGFP fluorescence dynamics with other known cell cycle markers.

*Figure 1 continued on next page*

*Figure 1 continued*

DOI: https://doi.org/10.7554/eLife.34025.007

**Figure supplement 6.** Cell segmentation and tracking pipeline.

DOI: https://doi.org/10.7554/eLife.34025.008

**Figure supplement 7.** Cell cycle selection procedure and its impact on cell cycle timings and measurements of compensatory growth.

DOI: https://doi.org/10.7554/eLife.34025.009

**Figure supplement 8.** Evolution of cell cycle duration as a function of replicative age.

DOI: https://doi.org/10.7554/eLife.34025.010

**Figure supplement 9.** Effect of multi-plane acquisition on cell cycle phase quantification.

DOI: https://doi.org/10.7554/eLife.34025.011

**Figure 1—video 1.** Cell cycle progression of wild-type cells.

DOI: https://doi.org/10.7554/eLife.34025.012

*supplement 2A*), the NLS-GFP strain yielded a continuous increase in total fluorescence throughout the cell cycle, as expected with a constitutive marker.

Next, we investigated how the maturation time of the fluorescent reporter affects the ability to accurately monitor the burst in histone level during S-phase. For this, we followed the expression of a second histone H2B marker, *HTB2*-mCherry, over time. Importantly, only a linear ramp followed by the anaphase drop could be discerned, in striking difference with the pattern observed with sfGFP (compare *Figure 1—figure supplement 4A and B* with *Figure 1—figure supplement 4C and D*). A numerical model confirmed that this effect could be quantitatively explained by the much longer maturation time of mCherry (~45 min [*Charvin et al., 2008*]) compared with sfGFP (~5 min [*Pédelacq et al., 2006*]), which blurs the apparent dynamics of histone synthesis (*Figure 1—figure supplement 4E–H*).

Although histone levels monitoring provides the timings of S phase and nuclear division, cytokinesis cannot be timed and, therefore, the duration of G1 cannot be deduced. To circumvent this problem, we used the septin subunit Cdc10-mCherry fusion as an additional cytokinesis marker (*Figure 1—figure supplement 5A–C*). We measured that cytokinesis (the sudden drop in Cdc10-mCherry fluorescence) and nuclear division were tightly correlated (Pearson coefficient 0.94), with a median offset of 5.6 ± 0.4 min between both events (*Figure 1—figure supplement 5D* and *Figure 1—figure supplement 5E*). Therefore, for the sake of simplicity, we chose to ignore cell-to-cell variability in this part of the cycle and, in the rest of the paper, we arbitrarily defined cell cytokinesis as an event occurring 5.6 min after the end of anaphase. We could not exclude the possibility that this procedure would introduce artefacts regarding measurements of G1 duration in mutants with cytokinesis defects. Yet, none of the reported mutants reported below, with the exception *cdh*1 (*Tully et al., 2009*), have been described to affect cell cytokinesis.

Similarly, we used the Whi5-mCherry fusion protein to assess the coordination between cell cycle Start (as defined by nuclear exit of the transcriptional repressor Whi5) and the onset of histone synthesis (Fig. *Figure 1—figure supplement 5F–H*). Start consistently occurred before the onset of histone synthesis (Fig. *Figure 1—figure supplement 5I–J*), which was expected because *HTB2* expression is controlled by the G1/S-specific transcription factors SBF/MBF. Taken together, these results confirmed the tight coordination between cell cycle progression and our measurements of the dynamics of histone expression.

To extend this preliminary analysis, we developed custom MATLAB software *Autotrack* to automate the processes of cell and nucleus contour segmentation, cell tracking, histone content measurement, and mother/daughter parentage determination (*Figure 1—figure supplement 6* and Supporting Information). We then used a piecewise linear model to identify the histone synthesis plateaus and ramp in the raw data, which allowed us to extract four intervals per cell cycle (*Figure 1B–C* and *Figure 1—video 1*): G1 (plateau), S (linear ramp), G2/M (plateau preceding anaphase), and the interval between anaphase onset and cytokinesis (referred to as 'Ana'), taking into account our hypothesis that the period between the end of anaphase and cytokinesis was constant, as mentioned above.

Using this method, we extracted the duration of cell cycle phases for up to ~500 cells in each of the eight cavities in each independent chamber. By pooling 17 replicate experiments, we collected ~26,900 cell cycles for WT cells (*Figure 1C*) of which 63% passed our quality control

procedure aimed at discarding cells with segmentation/tracking or data fitting issues (see Supporting Information and *Figure 1—figure supplement 7*). To decrease the rate of cell rejection due to noise in histone level signals, we tested multi-z-stack acquisition for HTB2-sfGFP fluorescence. However, this only marginally improved the signal to noise ratio (*Figure 1—figure supplement 8A–C*) while greatly affecting the cell cycle duration likely due to photo-damage (p<0.001, *Figure 1—figure supplement 8D*). Therefore, we retained the single plane acquisition method.

Using this analysis, we found that the cell cycle durations for WT cells were in good agreement with data obtained using other markers or methodologies. Thus, the durations for mothers and daughters, respectively, were: G1 (19.0 ± 0.1 and 45.5 ± 0.3 min) (*Di Talia et al., 2007*), S (29.5 ± 0.1 and 36.7 ± 0.2 min) (*Magiera et al., 2014*), G2/M (15.7 ± 0.1 and 14.7 ± 0.1 min), and Ana (11.4 ± 0.1 and 12.2 ± 0.1 min), *Figure 1D* and *supplementary file 2*. Importantly, the large sample size allowed us to identify statistically significant differences in these intervals. For instance, the S phase was 7.2 min longer in daughters compared with young mother cells (p<0.001; *Figure 1D*). In addition, this interval converged toward an asymptotic value over several divisions following cell birth (*Figure 1—figure supplement 9*). This contrasts with G1 duration, which decreased abruptly when daughters (replicative age 0, *Figure 1—figure supplement 9*) became mother cells (replicative age >0, *Figure 1—figure supplement 9*) in their subsequent division, and G2/M, the duration of which is quite independent of the replicative age of the cells. This phenomenon explains the previously reported (*Cookson et al., 2005*) *progressive* shortening of the cell cycle duration with the replicative age of the cell (see also *Figure 1—figure supplement 9*). Also, our result recapitulate the systematic shorter S/G2/M interval in mother cells compared to daughters that was recently measured(*Mayhew et al., 2017*). Over, these findings illustrate the power of our technique to quantitatively measure the temporal distribution of cell cycle intervals.

## Effects of environmental and genetic perturbation on the distribution of cell cycle phase durations

Because our methodology allowed us to detect even minor differences in cell cycle duration, we sought to validate its robustness by measuring the timing of cell cycle phases following perturbation by diverse environmental and genetic perturbations approaches that have been extensively studied using other techniques.

First, we asked whether our methodology could capture the lengthening of the S phase induced by hydroxyurea (HU), which inhibits DNA replication. As expected, we observed a progressive HU concentration-dependent increase in S phase duration in both mother and daughter cells, from an average of 27.1 ± 0.4 min and 35.1 ± 0.8 min at 0 mM to 48.0 ± 1.4 and 43.9 ± 1.4 min at 50 mM HU, respectively (p<0.001, *Figure 2A*). This prolongation of S phase was accompanied by a parallel doubling in G2/M duration (from ~15 min to ~30 min in both mothers and daughters, p<0.001, *Figure 2A*), which is likely due to the activation of the checkpoint that follows DNA damage (*Weinert et al., 1994*). Interestingly, mothers, but not daughters, exposed to HU experienced a dose-dependent slowing of the entire cell cycle, due to an apparent compensatory decline in G1 duration in the daughters (*Figure 2A*).

We next measured the duration of cell cycle phases in cells carrying mutations in important regulators of G1, S, or G2/M phases (*Figure 2B–D* and *supplementary file 2*). First, we confirmed that *sic1* and *whi5* (G1/S transition repressors) daughter cells underwent premature entry into S phase (i. e., shorter G1 duration) compared with WT cells (*Soifer and Barkai, 2014*), which was accompanied by a compensatory increase in G2/M duration (*Figure 2B*) (*Soifer and Barkai, 2014*). Conversely, mutation of G1/S transition activator *BCK2* (*Soifer and Barkai, 2014*), but not *CLN3*, caused a small delay in G1 of daughter cells. A slight decrease in G2/M duration was also observed in both *BCK2* and *CLN3* mutants compared with WT cells (*Figure 2B* and and *supplementary file 2*).

Next, we monitored the effect of mutations in genes involved in several biochemical pathways related to S-phase (Mrc1, Clb5, Dpb3, Rad27, Dia2). These mutations had previously been shown to induce an abnormally long S-phase interval using a cytometry assay (*Koren et al., 2010*). Our results confirmed this observation in all mutants, and, with the exception of *dia2*, the ordering of mutants according to S-phase duration was similar to Koren *et al.* (*Figure 2C*)(*Koren et al., 2010*). In addition, most of these mutants also displayed a longer G2/M phase, similar to the effect of HU treatment, with the exception of *dpb3* cells (*Figure 2A and C*). This result suggests that the increased G2/M duration observed following HU treatment and in most S phase mutants is likely to be

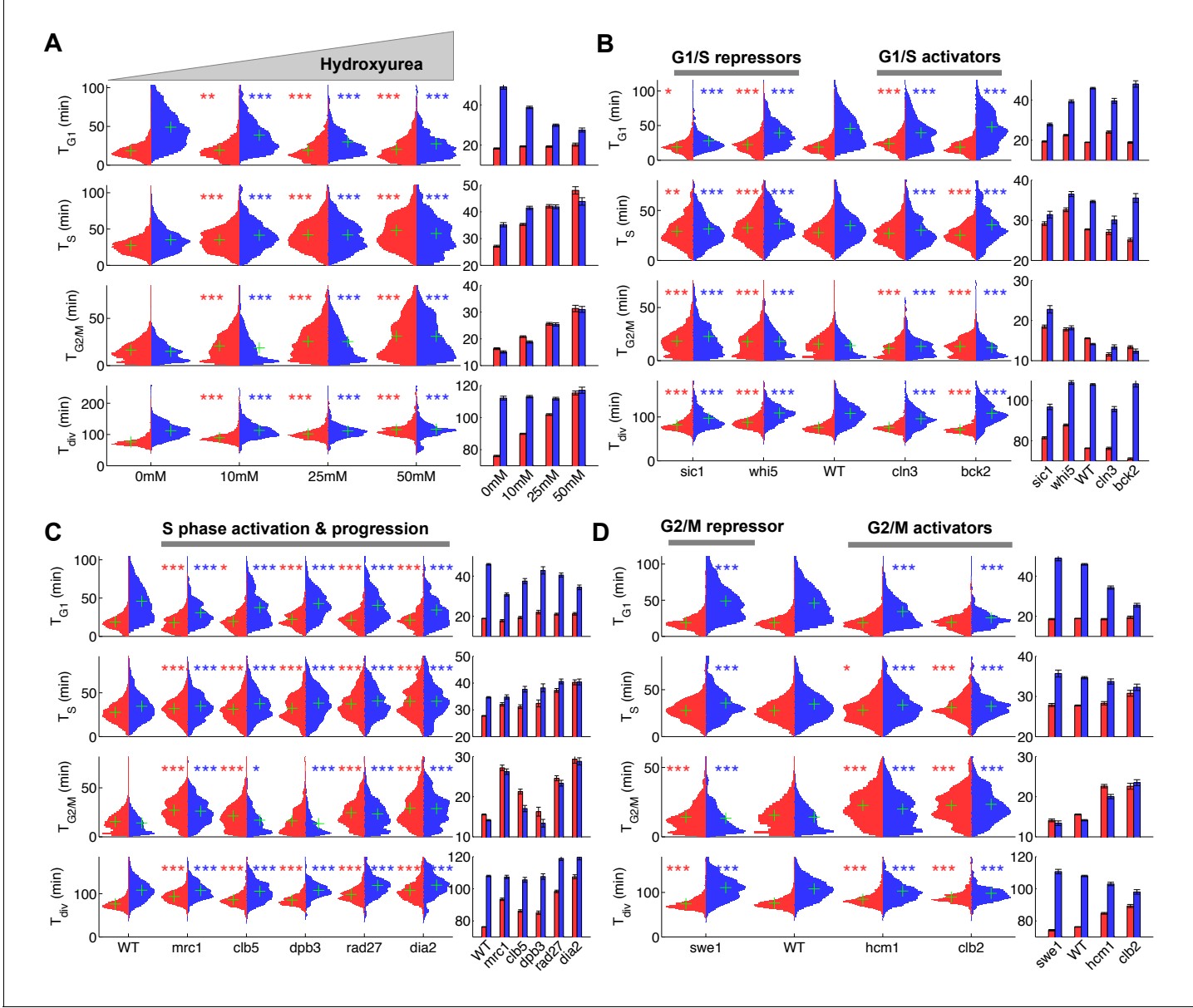

**Figure 2.** Duration of cell cycle phases in hydroxyurea-treated wild-type cells and in cell cycle mutant strains. (A) Left: Violin plots of the distribution of duration of G1, S, and G2/M phases and total division time ($T_{div}$) for mother (red) and daughter (blue) wild-type (WT) cells at the indicated hydroxyurea concentration. Green crosses indicate the distribution median. Star symbols indicate results of a Kolmogorov–Smirnov test: *p<0.05, **p<0.01, ***p<0.001 vs. 0 mM control. Right: Means of the distributions shown on the left. Error bars display standard error on mean (B–D) Same representation as in (A) analyzing various cell cycle mutants (untreated), using the WT strain as a reference for statistical tests.

DOI: https://doi.org/10.7554/eLife.34025.013

biological in origin rather than an artefact of our methodology. In this regard, a similar delay in G2/M progression was previously reported in *dia2*, *mrc1*, and *rad27* mutants, but not in *dpb3* mutants (*Koren et al., 2010*).

Lastly, we measured the cell cycle duration in mutants of G2/M progression. We found that deletion of *SWE1*, a kinase that inhibits Cyclin B/Cdk activity and therefore prevents a premature onset of anaphase, did barely affect G2/M duration, yet induced a slight increase in G1 duration of daughter cells, as previously observed (*Harvey and Kellogg, 2003*). However, mutants defective in Hcm1, a forkhead transcription factor that regulates late S phase genes, or Clb2, one of the main mitotic

cyclins, both displayed longer G2/M phases (*Figure 2D*), in agreement with previous measurements (*Pramila et al., 2006*; *Surana et al., 1991*).

Collectively, these results obtained in various mutant backgrounds further establish proof-of-principle for our methodology, in which a single fluorescent marker enables simultaneous measurements of key events associated with cell cycle progression. In total, we monitored the dynamics of cell cycle progression of 22 mutants. The raw cell cycle data are available on a dedicated server (*Tassy and Charvin, 2018*) that allows detailed data exploration and on-the-fly statistical analyses (see Appendix 1).

## Control of the metaphase to anaphase transition via a Bud-specific size compensatory mechanism

Our ability to measure the duration of specific cell cycle phases provides a unique opportunity to investigate in detail the coordination of growth and division during each phase of the cell cycle. We extracted 15 variables (e.g., phase durations, bud/cell volumes and growth rates during unbudded and budded period; see Supporting Information) describing cell cycle progression in both mother and daughter cells. Cell volumes were computed from segmented cell contours assuming an ellipsoid model.

Using this dataset, we sought to identify novel compensatory effects reflecting the existence of size control mechanisms. For this, we systematically measured the Pearson correlation coefficient for all measured distributions of variables in mothers and daughters (*Figure 3A and B*). This analysis successfully confirmed classical results, such as the negative correlation between size at birth ($V_{birth}$) and the duration of G1 ($T_{G1}$) in daughter cells (yellow star in *Figure 3B*) (*Di Talia et al., 2007*), which is indicative of a G1 compensatory mechanism in small daughter cells, as well as the positive correlation between the growth rate in unbudded cells ($\mu_{unb.}$) and the cell volume the end of G1 ($V_{G1}$) in daughters cells (green star in *Figure 3B*)(*Ferrezuelo et al., 2012*).

However, this analysis also revealed that the duration of G2/M ($T_{G2/M}$) varies inversely with the volume of the bud at the end of S phase ($Vb_S$) in both mother and daughter cells, see magenta stars in *Figure 3B*. This suggest that, after reaching the end of S phase, cells experience a bud size–dependent delay before entering anaphase. However, G2/M and S phase durations also appear negatively correlated, therefore, an alternative explanation would be that a longer S-phase provides more time for the cells to accumulate B-type cyclin, thus leading to a quicker anaphase onset and a reduction of the measured G2/M interval. In this case, the negative correlation between $T_{G2/M}$ and $Vb_S$ may result from the longer S phase that leads to a larger bud size by the end of S phase, as indeed observed (see the positive correlation between $Vb_S$ and $T_S$ on *Figure 3A*). To discriminate between the direct versus indirect link between these two variables, we first checked that the Pearson correlation coefficient $\rho$ was more pronounced for $T_{G2/M}$ and $Vb_S$ than for $T_{G2/M}$ and $T_S$ ($\rho = -0.43$ and $\rho = -0.29$, respectively), thus arguing in favor of a direct size-dependent modulation of G2/M duration. Second, we used a Bayesian statistics approach to determine which model (direct versus indirect link) fits better the data (*Meilă and Jaakkola, 2006*). This analysis confirmed the causative link between $Vb_S$ and $T_{G2/M}$ (see supplemental information for details).

Interestingly, this phenomenon is in agreement with the bud morphogenesis checkpoint model, which proposed that bud growth perturbations lead to a cell cycle arrest that prevents a potentially deleterious premature onset of anaphase (*Harvey and Kellogg, 2003*). It also matches the conclusions of a recent theoretical model of cell cycle control, in which bud size control appears to play an important role for the overall cell size homeostasis (*Spiesser et al., 2015*), as well as a recent statistical analysis of the duration of the budded period versus cell size (*Mayhew et al., 2017*).

However, another study provided evidence that this control mechanism does not operate as a bud size controller during an unperturbed cell cycle (*McNulty and Lew, 2005*). Therefore, to characterize this potential G2 bud size control further, we sought to determine the magnitude of compensatory growth effects during this phase of the cell cycle, and to compare it to the one of other phases. To this end, we monitored variation in cell volume $\Delta V$ during G1, G2/M, and the complete cell cycle as a function of initial cell volume (*Figure 3C and D*), according to a methodology widely used in previous studies (*Jun and Taheri-Araghi, 2015*): for an ideal Sizer, the variation in cell volume is such that the final volume $V_f$ is constant, independently of the initial one $V_i$. In this case, since $\Delta V = V_f - V_i$, plotting $\Delta V$ versus $V_i$ yields a linear relationship with a slope $-1$. For an ideal Timer (in which the duration of the phase is constant), the slope becomes $+1$, assuming an exponential growth

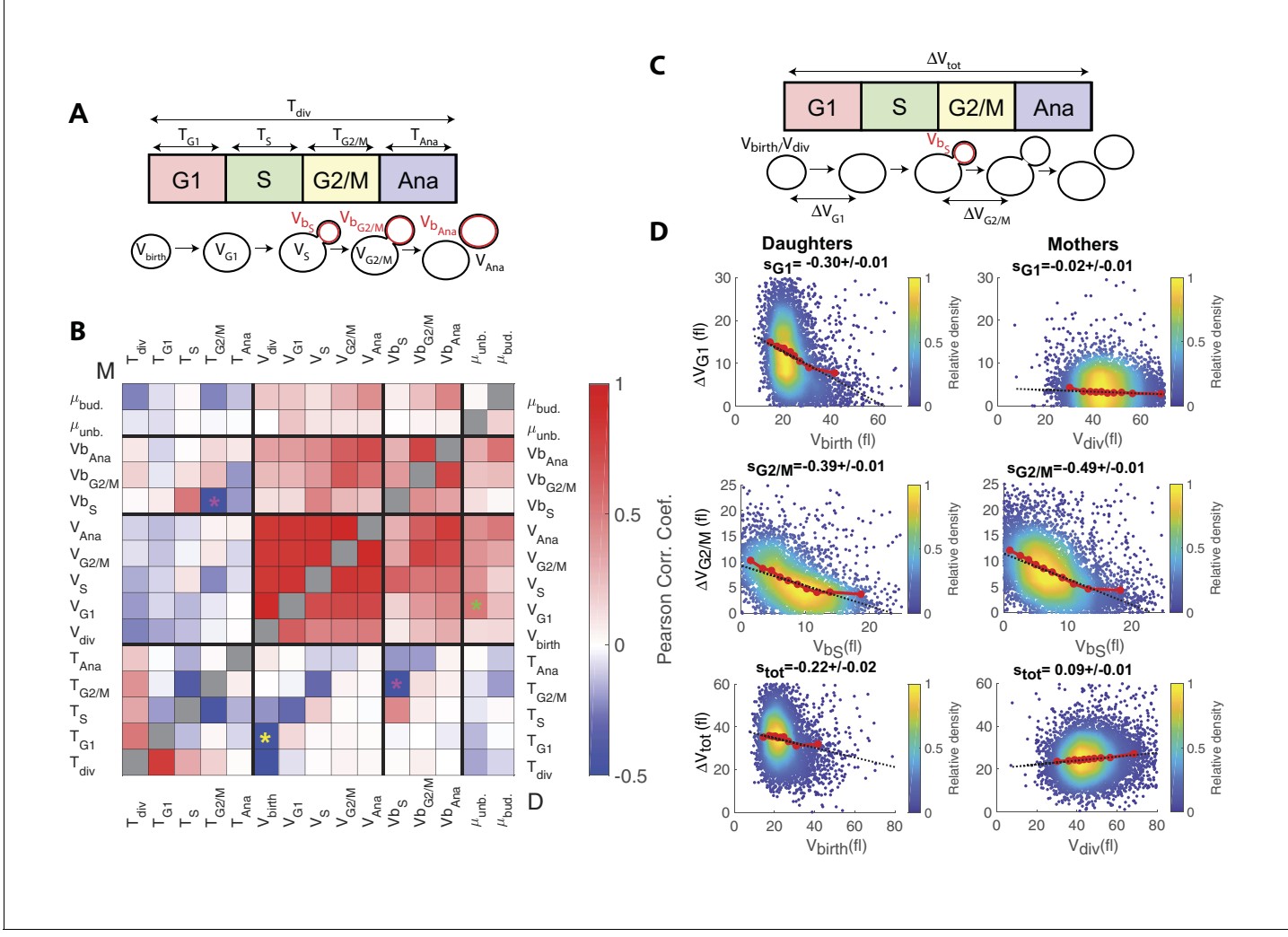

**Figure 3.** Identification and measurement of size compensation mechanisms in WT. (**A**) Top: Schematic of cell cycle phases and definitions of variables used in the correlogram in panel (**B**). (**B**) Correlogram that represents Pearson's correlation coefficient obtained from the scatter plot associated with the two variables. As indicated on the color scale, blue indicates a negative correlation and therefore highlights the presence of a potential compensatory mechanism, whereas red indicates a positive correlation. T indicates the duration (min) of each cell cycle phase; V and $V_b$ indicate the mother and bud volumes at each cell cycle phase, respectively. $\mu_{unb}$ and $\mu_{bud}$ are the linear growth rate during the unbudded and budded period of the cell cycle, respectively. Ana indicates anaphase to cytokinesis interval. M (top left triangle) and D (bottom right triangle) represent the analyses performed in mother and daughter cells, respectively. Colored asterisks indicate squares of specific interest (see Main text). (**C**) Schematic of cell cycle phases and definitions of variables used in the scatter plots below. (**D**) Scatter plots showing variations in mother/bud volumes at the indicated cell cycle stages. Color indicates point density, according to the indicated color code. Red line shows binning of the scatter plot along the x-axis. Dashed black line is a robust linear regression through the cloud of points, and the indicated slope (**s**) represents the strength of the size-compensation mechanism. Error bars represent a 95% confidence interval.

DOI: https://doi.org/10.7554/eLife.34025.014

The following figure supplement is available for figure 3:

**Figure supplement 1.** Comparison of size compensation effects in the budded part of the cell cycle.

DOI: https://doi.org/10.7554/eLife.34025.015

model, and a doubling of cell size during the considered interval(*Jun and Taheri-Araghi, 2015*). Last, an Adder is such that the amount of added volume is independent of the initial volume; in this case, the slope is 0. Therefore, measuring the slopes *s* of $\Delta V$ vs. $V_i$ plots provide a quantitative assessment of the magnitude of size compensation effects, as well as their deviation from theoretically ideal behaviors (i.e. Sizer, Adder and Timer).

The analysis captured the well-characterized daughter-specific Sizer in G1 (*Figure 3D*: slope $s_{G1}$ is negative in daughters, $-0.30 \pm 0.01$, but approaches zero in mothers, $-0.02 \pm 0.01$) (*Di Talia et al., 2007*). It also confirmed that the daughter and the mother cells behave as a weak Sizer and Adder, respectively, over the entire cell cycle ($s_{tot} = -0.22 \pm 0.02$ for daughters and $0.09 \pm 0.01$ for mothers) – in previous studies, the low absolute values of $s_{tot}$ showed that the budding yeast cell cycle behaves as an Adder, as observed in other unicellular organisms (*Jun and Taheri-Araghi, 2015*; *Soifer et al., 2016*).

In addition, the data clearly indicated the existence of a bud-specific size-compensatory growth in G2/M in both daughter and mother cells ($s_{G2/M} = -0.39 \pm 0.01$ and $-0.49 \pm 0.01$ in daughters and mothers, respectively, *Figure 3D*). Importantly, we found that the magnitude of the Sizer was strongly reduced when considering the total cell volume (rather than only the bud, see *Figure 3— figure supplement 1B*), or when measuring the magnitude of size compensation during the whole budded period of the cell cycle (*Figure 3—figure supplement 1C*). This very likely explains why G2/M size control has been largely ignored in budding yeast and has been considered as 'cryptic'. Instead, our measurements revealed that the G2/M bud size control is of comparable magnitude to the long-known size control in G1. Therefore, these results suggest that at least two mechanisms may act coordinately to ensure size homeostasis throughout the cell cycle (*Soifer and Barkai, 2014*).

## Impaired size control in mutants of cyclin B regulation and function

To better characterize the molecular basis of size compensation effects throughout the cell cycle, we measured the magnitude of compensatory growth for daughter cells (using ΔV vs. V plots, as in *Figure 3C*) in a subgroup of the cell cycle progression mutants examined earlier (*Figure 2*). We found that deletion of the repressor of G1/S cyclin Whi5 decreased the magnitude of G1 control ($s_{G1} = -0.17 \pm 0.02$) but compensated with a slight increase in G2/M control ($s_{G2/M} = -0.46 \pm 0.02$, *Figure 4*). However, these changes were relatively modest, and the overall size-compensation slope was similar in *whi5* and in WT cells ($s_{tot} = -0.26 \pm 0.03$ and $s_{tot} = -0.22 \pm 0.02$, respectively; *Figure 4*). Deletion of other G1/S regulators (*cln1, cln2, cln3, swi4*) lead to similar conclusions. However, G1 size compensation was slightly improved by deletion of the activator of G1/S transition *BCK2* ($s_{G1} = -0.34 \pm 0.04$), and the overall compensatory growth was stronger than in WT cells ($s_{tot} = -0.46 \pm 0.06$).

In striking contrast to these G1/S regulators, deletion of other cell cycle control genes related to the control of B-type cyclin function, such as *sic1, swe1, clb5,* and *clb2*, induced a much larger decrease in size compensation in both G1 and (with the exception of *clb2*) G2/M phases, as well as the overall compensatory growth (*Figure 4*). For instance, loss of Swe1, which inhibits Cyclin B-Cdk activity and regulates the onset of anaphase, leads to a slightly Timer-like behavior ($s_{tot} = 0.15 \pm 0.05$), in which both G1 ($s_{G1} = -0.12 \pm 0.03$) (*Soifer and Barkai, 2014*) and G2/M ($s_{G2/M} = -0.15 \pm 0.04$) size compensation were largely abolished. Taken together, these data indicate that the compensatory mechanisms ensuring the control of cell size were strongly affected in mutants linked to the regulation cyclin B-Cdk activity but, unexpectedly, only marginally impaired in mutants of the G1/S control network.

## Effective cell size homeostasis during cell cycle progression

To determine how size G1 and G2/M compensation effects actually impact size homeostasis, we quantified cell size variability during cell cycle progression. We measured the coefficient of variation (CV) of the distributions of cell/bud volumes at various points in the cell cycle from bud emergence to the next division of the resulting daughter cell (*Figure 5A*). We found that the CV gradually decreased as a function of cell cycle progression and cell size, roughly following a square-root dependency: $CV = F^{1/2} / \langle V \rangle^{1/2}$, where F is a constant and $\langle V \rangle$ is the mean cell/and or bud volume at a given point in the cell cycle. This scaling relationship between CV and cell size is to be expected, according to the Central Limit Theorem, assuming that cell growth is the sum of elementary stochastic processes: growth fluctuations tend to average out in larger cell compartments compared to smaller ones. Therefore, to characterize the intrinsic variability in cell size during cell cycle progression and to facilitate comparison among mutants of diverse sizes, we evaluated F (known as the Fano factor [*Fano, 1947*]), rather than the CV, because F provides a size-independent measurement

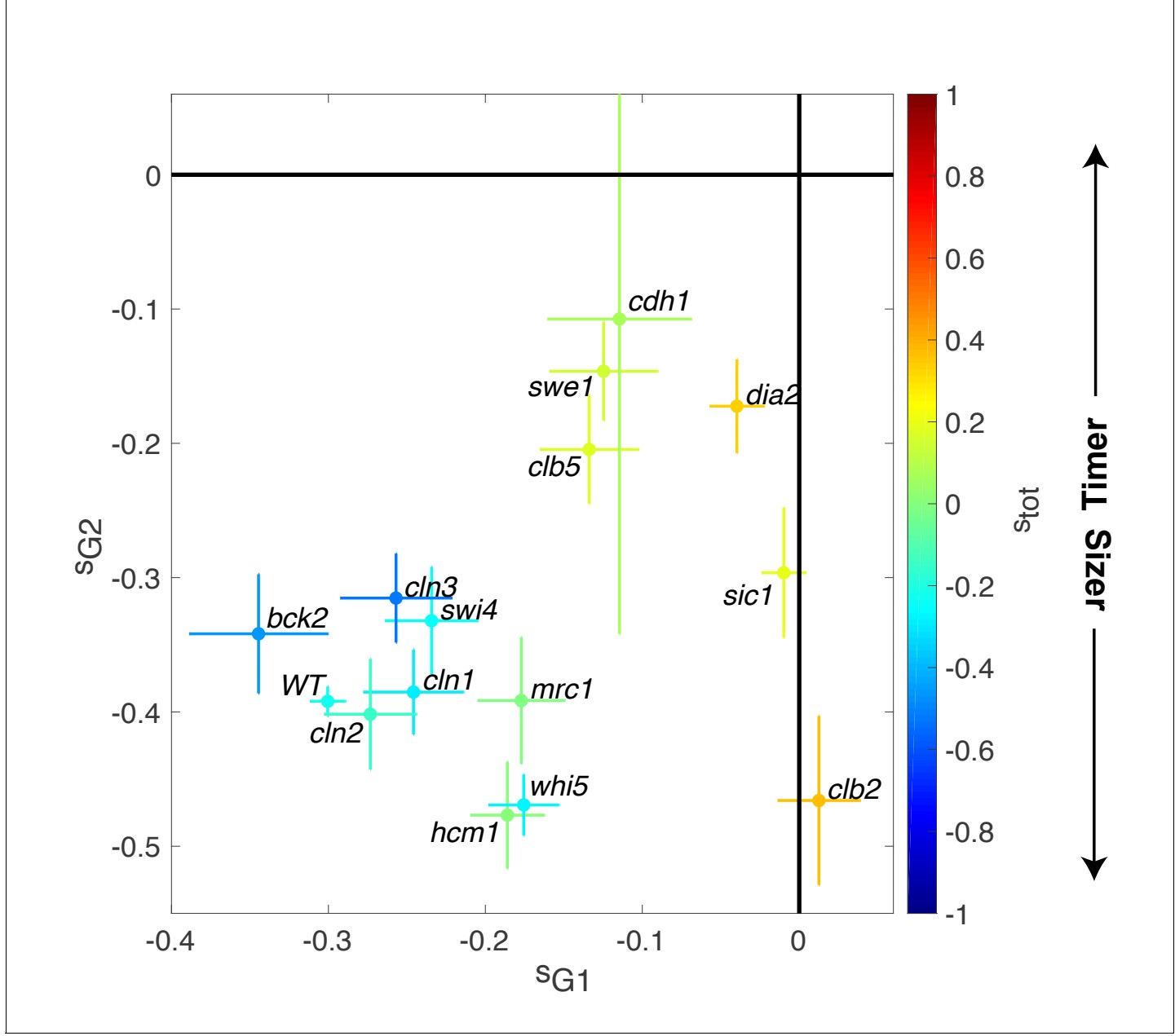

**Figure 4.** Magnitude of size compensation effects in cell cycle mutants. Strength of size compensation during G1, G2/M, and the entire cell cycle in the indicated mutant backgrounds, calculated as described in *Figure 3D*. The cross color indicates the overall compensation size during the entire cell cycle, as indicated by the color scale. Values of −1 and +1 correspond to an ideal Sizer and Timer, respectively. Error bars represent a 95% confidence interval obtained from robust linear regression.

DOI: https://doi.org/10.7554/eLife.34025.016

of noise in cell size during the cell cycle (F is sometimes referred to as noise strength [*Raser and O'Shea, 2004*]).

As expected, the Fano factor was much more stable than the CV during cell cycle progression of WT cells (*Figure 5A*). Still, it displayed some notable variations around the mean at different points in the cell cycle: specifically, F decreased during G2/M and G1 phases, but increased during the rest of the cell cycle, especially during S phase (*Figure 5A*). The Fano factor associated with cell size was much higher than the noise due to segmentation errors (see Material and methods for detail), thus ruling out the possibility that measurements of cell size variability might be dominated by

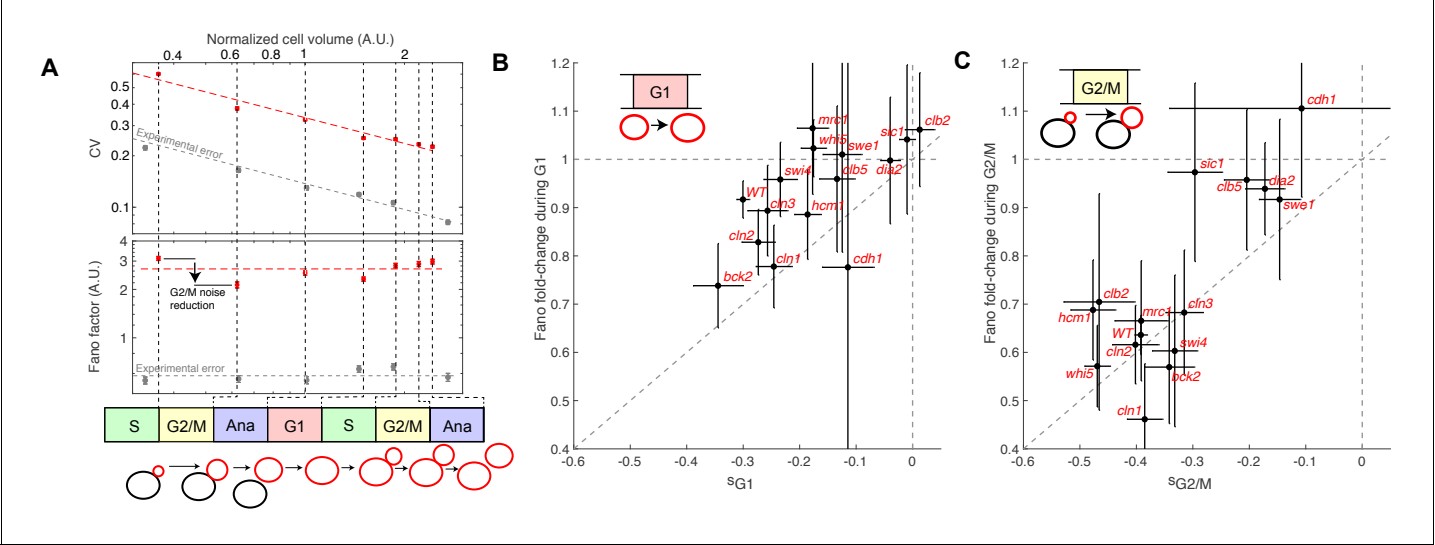

**Figure 5.** Evolution of cell size variability during cell cycle progression. (**A**) Measurements of cell and/or bud size variability as a function of the mean cell and/or bud volume (in logarithmic scale) during cell cycle progression. Each data point (red symbols) corresponds to a measurement at specific cell cycle phases, as indicated in the bottom schematic. Top panel represents the coefficient of variation (CV, red symbols) in logarithmic scale. Error bars represent the statistical error, which was estimated by bootstrap analysis. The red dashed line shows the best fit to a power law $CV=F^{1/2}/V^{1/2}$, where F is the Fano factor (see main text). The grey symbols represent an estimate of experimental error on cell volume measurement, as a function of cell size (see Materials and methods for detail). Error bars represent the standard error on mean measurement error (grey symbol, see Material and methods). Bottom panel shows the Fano factor (in logarithmic scale), as defined above (red symbols, see main text for detail). Error bars represent the statistical error, which was estimated by bootstrap analysis. The red horizontal dashed line represents the mean Fano factor during cell cycle progression. The grey symbols represent an estimate of noise associated with cell volume measurement, as a function of cell size (see Materials and methods for detail). Error bars represent the standard error on noise measurement (grey symbol). (**B**) Ratio of the Fano factors at the end and the beginning of the G1 phase for the indicated mutants as a function of the magnitude (slope) of the size-compensation mechanism, as defined in (**B**). Error on the mean Fano was calculated using a bootstrap test, whereas the error on slope was calculated using a robust linear regression procedure (95% confidence interval). The dashed line has a slope one and coincides with the point (0;1). (**C**) Same as in (**B**), except applied to bud growth during the G2/M phase.

DOI: https://doi.org/10.7554/eLife.34025.017

experimental noise. Instead, our results support the hypothesis that cell size noise is clearly modulated during cell cycle progression. To check this further, we asked whether the magnitude of the decrease in Fano factor at specific cell cycle phases was consistent with that of the compensatory growth. For this, we plotted the fold-change in Fano during G1 (*Figure 5B*) and G2/M (*Figure 5C*) for each of the cell cycle mutants. Importantly, we observed that mutants with strong size compensation effects (i.e. with a negative slope) displayed larger reductions in Fano factor in both G1 and G2/M phases. Of interest, the reduction in Fano factor was larger in G2/M (fold-change ~0.65) than in G1 (~0.9) in WT cells (*Figure 5B and C*), confirming the importance of cell size control during G2/M. Also, this analysis clearly demonstrated that the magnitude of size compensation mechanisms directly influences size homeostasis in a cell cycle phase-specific manner.

## A linear map model linking size control efficiency to size homeostasis

Following the analysis of compensatory growth during G1 and G2/M, we wondered how the overall (i.e., during a full cell cycle) daughter cell size homeostasis was dependent on the overall size control ($s_{tot}$) in the cell cycle mutants. To explain the quantitative relationship between the magnitude of size control due to compensatory growth and actual variability in cell size, we turned to a noisy linear map of the cell cycle, which provides a simple model to couple phenomenological parameters that describe cell growth and division (*Tanouchi et al., 2015*). Under this assumption, the evolution of daughter cell volume $V_n$ at the beginning of the cell cycle $n$ can be given by (see Supporting Information for detail):

$$V_{n+1} = p\,V_n + (1-p)\,V_{eq} + \eta \tag{1}$$

with $p = r\,a$, where $a$ characterizes the efficiency of size control (which is directly related to the magnitude of size compensation, represented by the slope $s_{tot}$ measured in Figure 3B: $a = s_{tot} + 1$, see Supporting Information), $r$ is the fraction of volume going to the daughter cell at division (asymmetry factor, $0 < r < 1/2$), $V_{eq}$ is the volume of a daughter cell at equilibrium (Fig. 6A), and $\eta$ represents a Langevin noise, such that $<\eta> = 0$ and $<\eta^2> = $ constant. Under these assumptions, we demonstrated that the Fano factor is given by (see Supporting Information for detail):

$$Fano = \frac{<\eta^2>}{V_{eq}} \frac{1}{(1-p)(1+p)} \qquad (2)$$

This equation indicates that the variability in cell size depends on a, size-independent, intrinsic noise constant $\frac{<\eta^2>}{V_{eq}}$, which reflects the stochasticity of the growth process, and an effective size control parameter $p$ (with $0 < p < 1$). Notably, this model predicts a non-linearity in size variability as a function of size control parameters, and has two interesting limit cases: for a perfect Sizer ($s_{tot} = -1$), $p$ equals 0, therefore the Fano factor equals the intrinsic noise associated with the growth process, given by $<\eta^2>/V_{eq}$. In contrast, for a perfect Timer ($s_{tot} = 1$), and assuming symmetrical division of mother and daughters ($r = \frac{1}{2}$), $p$ equals 1 and thus there is a divergence in Fano factor, leading to a complete loss of size homeostasis. In the case of budding yeast, which divides asymmetrically ($r < \frac{1}{2}$), such extreme case is impossible. In other words, even with a perfect Timer, asymmetrical division is sufficient to limit cell size variability.

To check the validity of this description, we computed the average Fano factor during the cell cycle and calculated $p$ by robust linear regression of single-cell data in WT and mutants (*Figure 6A*). We observed large variations in Fano factor among the mutants, which appeared to be correlated with the size control parameter $p$ (Pearson correlation coefficient = 0.60, *Figure 6B*): overall, Sizers (i.e. with low values of $p$) tend to have less cell size noise than Timers (i.e. high values of $p$). Interestingly, mutants related to the G1/S network (*bck2, swi4, whi5, cln1*, with the exception of *cln2* and *cln3*) generally displayed a noise level comparable to WT and lower than did the mutants associated with the regulation of B-type cyclin function (*clb5, swe1, clb2, cdh1; Figure 6B*).

Fitting the model prediction to the experimental data (using a single parameter fit $<\eta^2>/V_{eq}$) yielded reasonable agreement, despite a large spread in the experimental data and the existence of an outlier, the *dia2* mutant, which failed to fit the model (*Figure 6B*). Therefore, this analysis revealed that the degree of size variability observed in this cohort of cell cycle mutants, associated with diverse roles in cell cycle progression, can be reasonably accounted for by a simple model in which there is a universal noise parameter that characterizes the stochasticity of the growth process, as well as a mutant-specific parameter associated with size control. The deviation of experimental data from the predictions of the model are likely to originate, in part, from the simplistic assumption that size control is a homogenous process throughout the cell cycle, thus ignoring the contributions of size compensation mechanisms in specific phases. In addition, whereas the hypothesis of a linear map was correct for some strains (e.g. WT in *Figure 6A*), some deviations were observed in others (i.e. multiple slopes may be needed to describe the behavior of *bck2* in *Figure 6A*).

Interestingly, the parameter $p$ is related to the rate $\lambda$ of convergence of the linear map, which sets the time it takes for a chain of daughter cells to return to equilibrium following a fluctuation in cell size:

$$p = \exp(-\lambda <T_{div}>) \qquad (3)$$

where $<T_{div}>$ is the average generation time of a specific mutant. By computing $\lambda$ from $p$ and $<T_{div}>$, we identified a large spread in convergence rates, ranging from $1.9 \times 10^{-3}$ min$^{-1}$ to $5.7 \times 10^{-3}$ min$^{-1}$ (*Figure 6C*). By selecting the fraction of daughter cells that deviated significantly (larger or smaller) from the equilibrium size at birth and tracking the average size of consecutive daughters, we confirmed that convergence to equilibrium was impaired in the *swe1* mutant but slightly improved in the *bck2* mutant (yet only for large cells, presumably because the slope $p$ seems different for small versus large *bck2* cells on *Figure 6A*, see above) compared with WT cells (*Figure 6D*). Therefore, this analysis revealed that mutations of cell cycle regulators not only modify the average duration of specific phases but also control the timescale of size fluctuations and hence the robustness of the cell cycle orbit.

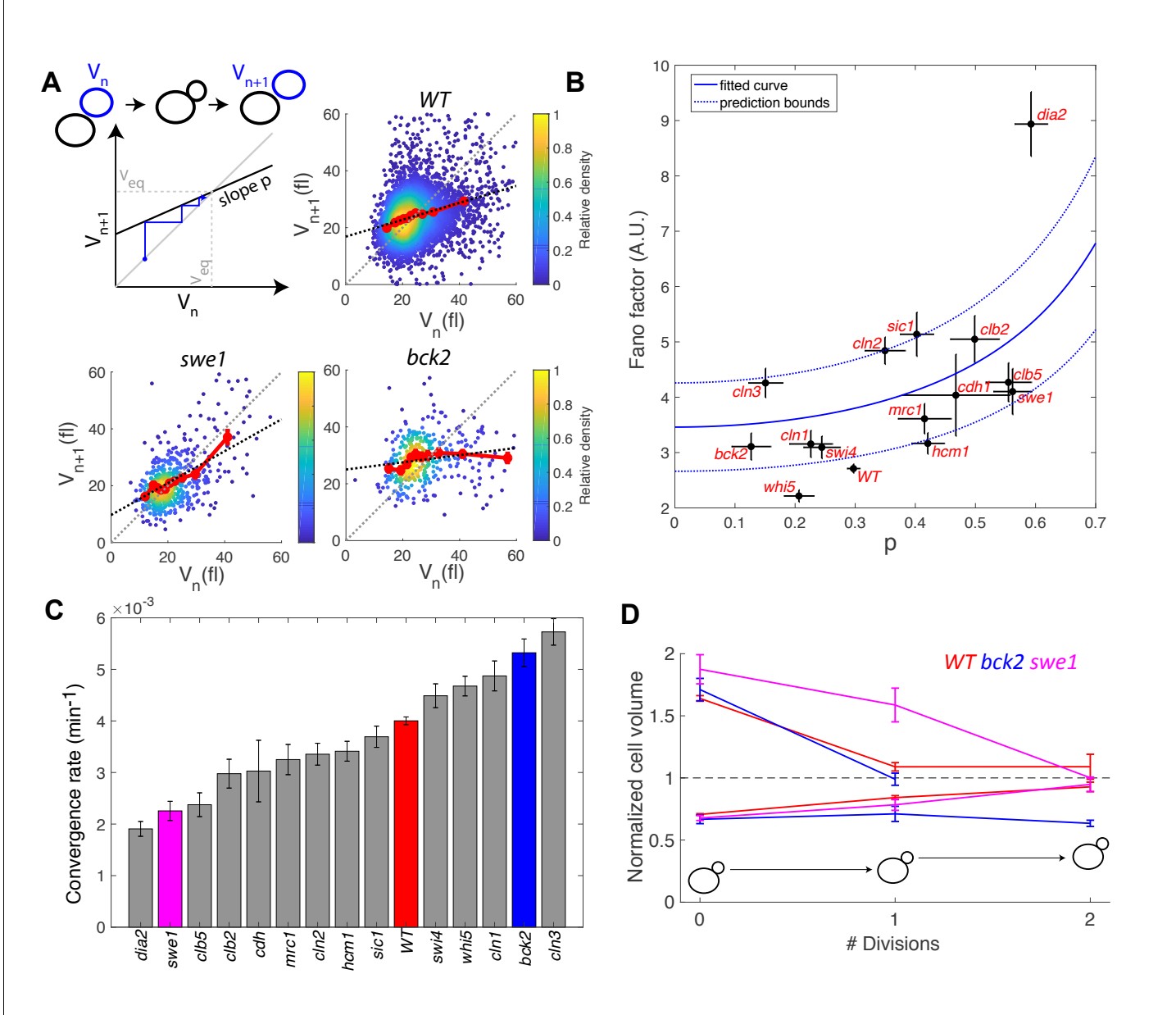

**Figure 6.** Return map analysis linking cell size noise to the magnitude of the size-compensation mechanism. (A) Top left: Illustration of the return map model, showing the successive iterations of daughter cell size at birth $V_n$. The size-compensation mechanism can be described by three parameters: steady-state volume $V_{eq}$, size-compensation strength p, and noise η. The three return maps of experimental data were obtained with wild-type (WT), *swe1*, and *bck2* daughter cells. Color indicates point density. Red line shows binning of the scatter plot along the x-axis. Dashed black line is a linear regression through the cloud of points. The gray dashed line is the diagonal. (B) Average Fano factor during the entire cell cycle as a function of the experimentally measured size-compensation strength p. The black points and bars are the mean ± SEM of the WT cells and indicated mutants. The blue line shows the single parameter fit to the model (see text), yielding the intrinsic noise of the growth process $\frac{<\eta^2>}{V_{eq}} = 3.5\pm 0.5$, with 99% confidence intervals indicated by the dashed blue lines. (C) Rate of convergence to an equilibrium size for each strain listed in order, based on *Equation 3* in the main text. Mean ± SEM. (D) Normalized size of successive daughter cells for WT cells and swe1 and *bck2* mutants, starting from cells that deviate by more than 50% or less than 30% of the equilibrium cell size (indicated by the black dashed line).
DOI: https://doi.org/10.7554/eLife.34025.018

## Discussion

In this paper, we have described a new technique to monitor the duration of successive phases of the cell cycle based on quantification of histone level dynamics in individual growing yeast cells. Most previous single cell analyses tracked cell cycle progression through budding events, ignoring the details of S/G2/M phase events. Our methodology overcomes this limitation and offers new perspectives on the quantification of temporally controlled events in individual cells, such as the coordination between DNA replication and mitosis. Notably, unlike other markers of cell cycle progression (*Sakaue-Sawano et al., 2008*), our technique is based on a single fluorescent marker, thereby enabling correlative measurements to be made using additional spectrally independent markers.

The large throughput of the image acquisition/processing pipeline developed in our study provides the opportunity to detect mild yet meaningful phase duration phenotypes that were not detected in previous analyses. The discovery that cells replicate their DNA more rapidly with increasing replicative age is a good example of this ability to resolve small differences in cell cycle timing, although this observation needs to be confirmed using complimentary techniques. Since we could not exhaustively analyze the large datasets generated in this study within the scope of this article, we have created a dedicated server to allow further statistical analyses of cell cycle variables in individual cells.

The main interest of our methodology was to enable the identification of size compensation effects throughout the cell cycle in an unbiased approach, and the possibility to assess their role in the establishment of size homeostasis in a quantitative manner. Building on previous studies (*Harvey and Kellogg, 2003*), our work now clearly establishes the link between bud growth and cell cycle progression through G2, and reveals that the magnitude of size compensation is comparable to the well-known G1 size control. Although budding and DNA replication are triggered concomitantly by the activation of the G1/S regulon, the noise in bud size that is observed at the end of S phase suggests that these two processes appear to be largely uncoordinated. Therefore, the function of bud size control during G2/M may be to prevent the potentially deleterious onset of anaphase in small-budded cells following DNA replication. Importantly, this finding challenges the idea of a 'cryptic'-type G2 size control in budding yeast, which would only be observed upon appropriate environmental or genetic perturbations. Instead, it supports the hypothesis of universal size control mechanisms across eukaryotes, like fission yeast, in which a G2 size control has long been established (*Fantes, 1977*).

Beyond previous work focusing on the identification of G1-specific size compensation regulators (*Soifer and Barkai, 2014*), our analysis in mutants broadens our understanding of how the emergence of size homeostasis is connected to the cell cycle control network. Unexpectedly, we found that mutations in activators or repressors of G1 progression had only marginal effects on the overall noise in cell size. In particular, while mutating Whi5 slightly decreased G1 compensatory growth and reinforced G2/M size control, the overall size homeostasis was quite preserved in this mutant. Interestingly, we observed a slight but significant increase in G1 size compensation effects in the *bck2* mutant compared with WT cells, indicating that this phenotype is genetically tunable in both directions.

In contrast, we found that mutations of regulators of cyclin B-Cdk activity had a more pronounced effect on cell size homeostasis: G2/M size compensation was largely abolished in the *swe1* mutant (*Harvey and Kellogg, 2003*), as well as in *sic1* and *cdh1* mutants. Strikingly, all of these mutations also reduced the magnitude of the G1 size control. With a few notable exceptions (*hcm1* or *clb2*), the fact that the magnitude of G1 and G2/M size compensations are somewhat coupled across these mutant strains suggests that enforcing a clear switch in B-type cyclin-Cdk activity between the low (S/G2/early M) and high (late M/G1) APC activity regimes is critical for cell size homeostasis. Further modeling of cell cycle dynamics using a detailed molecular description will be important to clarify this point (*Tyson and Novak, 2011*).

A novel feature of our linear map model is the proposed general formula linking the efficiency of cell size control to the noise in cell size, which is reasonably well supported by the experimental data obtained in various mutant backgrounds (*Figure 6B*). This model predicts a divergence in Fano factor when the behavior of the cell approaches that of an a ideal Timer (*Taheri-Araghi et al., 2015*). We speculate that the deletion of some cell cycle genes may render cells inviable, not to loss of an essential biochemical function, but rather to complete loss of size homeostasis, thus impairing the

robustness of the cell cycle oscillation. Conversely, even a cell cycle mutant in which $s_{tot}$ = +1 (i.e., a perfect Timer according to the previous definition [*Jun and Taheri-Araghi, 2015*; *Tanouchi et al., 2015*]), should be able to control its size if dividing asymmetrically (since $p = r(s_{tot}+1) < 1$ when $r < ½$). Therefore, asymmetric division can be regarded as an additional stabilizer of cell cycle that limits cell size variability.

In conclusion, our study, in which cell cycle progression was monitored with unprecedented accuracy in yeast, demonstrates that size homeostasis does not originate from a G1-specific mechanism, but is likely to be an emergent property resulting from the integration of at least two mechanisms that coordinate cell growth with division. Our analysis specifically highlights the role of bud size control in limiting cell-to-cell variability, which is presumably connected to the role played by B-type cyclins in size homeostasis, as identified here. Additional studies linking further experimental datasets to computational analyses (*Tyson and Novák, 2015*) will be instrumental in deciphering how individual components are integrated to ensure size homeostasis throughout the cell cycle.

## Materials and methods

### Strain construction

All strains were congenic to S288C unless specified otherwise and were constructed following standard genetic techniques. A detailed list of strains is provided as a *supplementary file 1*. HTB2-sfGFP fusion protein was generated by classical PCR-mediated genome editing. Mutant strains were obtained from the deletion collection of non-essential genes. In the list of constructed strains, we noticed that the *cdh1Δ HTB2-sfGFP* strains were quite unstable and yielded a large fraction of dead cells as well as large multinucleated cells that retained a fast division time and eventually outgrew the rest of the population. Indeed, the *cdh1Δ* mutation has previously been described to induce genomic instabilities (e.g. chromosome loss, etc..)(*Ross and Cohen-Fix, 2003*). We hypothesize that introducing the histone marker in this background exacerbates this phenotype. To circumvent this issue, we used freshly thawed cells from frozen stock.

### Microfabrication and microfluidics setup

Microfluidic chips were designed and made using standard techniques as previously described (*Goulev et al., 2017*). The microfluidic devices, which feature eight independent channels, each consisting of 8 chambers, allow parallel monitoring of 8 genetic backgrounds in the same time-lapse assay. The microfluidic master was made using a standard SU-8 lithography process at the ST-NANO facility of the IPCMS (Strasbourg, France). CAD files and detailed dimensions of the chip are available on the metafluidics open repository: https://metafluidics.org/devices/yeast-high-throughput-culture-device-with-8-independent-flow-chambers/. The micro-channels were cast by curing PDMS (Sylgard 184, 10:1 mixing ratio) and then covalently bound to a 24 × 50 mm coverslip using plasma surface activation (Diener, Germany). Chips were then baked for 1 hr at 70°C to improve the sealing between PDMS and glass. Microfluidic chips were connected using Tygon tubing and media flows were driven by a peristaltic pump (Ismatec, Switzerland) with a 30 µL/min flow rate.

### Live imaging of yeast

Strains were cultured overnight in synthetic complete medium with glucose and all amino acids. The next morning, the cultures were diluted and allowed to grow until optical density at 660 nm reached 0.2–0.5. Each strain was then loaded into independent chambers chosen at random to avoid potential systematic bias in measurements. For each strain, at least two fields of view were recorded during the time-lapse acquisition interval. Each assay included a WT strain as a control. Mutants were analyzed in at least three independent assays. In total, we collected at least 2000 cell cycles per mutant ~25,000 cell cycles for the WT strain.

Cells were imaged every 3 min using an automated inverted microscope (Nikon TI, Nikon, Japan) with a 60 × phase contrast objective and a sCMOS camera (Hamamatsu Orca Flash 4.0, Japan) driven by Nikon Software (NIS). Constant focus was maintained using a Perfect Focus system. Fluorophore excitation was performed using LED light (Lumencor X1) and appropriate filter sets. The cells were allowed to grow for up to 10 hr in the device, yielding about 500 cells per field of view by the end of the assay.

## Image processing

Raw proprietary Nikon files (.nd2 format) were converted using Bio-format and bftools packages into MATLAB-compatible lossless jpeg files. We developed custom software (Autotrack; see Fig. *Figure 1—figure supplement 2* and Supplemental Information for details) to: (1) segment and track individual cells in yeast microcolonies, (2) quantify HTB2-sfGFP (or mCherry) levels in individual cells and extract individual cell cycles, (3) determine the duration of individual cell cycle phases, and (4) discard outliers based on specific criteria (detailed in Supplemental Information). Cell volumes were calculated from segmented cell contours assuming an ellipsoid model.

## Data processing

Linear regression was performed with a robust regression procedure (*robustfit*) function in Matlab) using a weighting function to limit the impact of potential outliers (Error estimates correspond to a 95% confidence interval). Volume measurement errors reported in *Figure 5A* were estimated by comparing the volume obtained from automated cell segmentation to a manual ground truth segmentation performed over more than 500 cells of various sizes.

## Dataset management and online data publishing

All variables extracted during image processing were stored in a mutant-specific database designed to allow straightforward analysis using custom MATLAB software, as described in the Supporting Information. We developed a web application, Yeast Cycle Dynamics, that allows custom statistical analysis of extracted cell cycle data for all mutants in this study. In addition, raw data showing histone levels and cell size as a function of time for all cells can be monitored (*Tassy and Charvin, 2018*). See Supplemental Information for details.

## Acknowledgements

We are very grateful to Damien Coudreuse for extensive discussion and feedbacks on the manuscript, Manuel Mendoza and Etienne Schwob for insightful discussions, as well as Sophie Quintin and the Charvin lab for careful reading of the manuscript. We are very grateful to Felix Jonas for pointing an experimental error in the preprint version of this manuscript. We thank Sandrine Morlot for help with microscopy. We thank Michaël Knop for sharing plasmids and Joseph Schacherer for the gift of the cell cycle mutants. This work was supported by the ATIP-Avenir program (GC), a grant from the Fondation pour la Recherche Médicale (GC), and by grant ANR-10-LABX-0030-INRT, a French State fund managed by the Agence Nationale de la Recherche under the frame program Investissements d'Avenir ANR-10-IDEX-0002–02.

## Additional information

### Funding

| Funder | Grant reference number | Author |
| --- | --- | --- |
| Centre National de la Recherche Scientifique | ATIP–Avenir | Gilles Charvin |
| Agence Nationale de la Recherche | ANR-10-LABX-0030-INRT | Gilles Charvin |
| Agence Nationale de la Recherche | ANR-10-IDEX-0002-02 | Gilles Charvin |
| Institut National de la Santé et de la Recherche Médicale | ATIP–Avenir | Gilles Charvin |
| Fondation pour la Recherche Médicale | | Gilles Charvin |

The funders had no role in study design, data collection and interpretation, or the decision to submit the work for publication.

## Author contributions
Cecilia Garmendia-Torres, Resources, Data curation, Investigation, Methodology, Writing—review and editing; Olivier Tassy, Software, Validation, Visualization; Audrey Matifas, Resources, Investigation, Methodology; Nacho Molina, Investigation; Gilles Charvin, Conceptualization, Software, Formal analysis, Supervision, Funding acquisition, Validation, Investigation, Methodology, Writing—original draft, Project administration, Writing—review and editing

## Author ORCIDs
Gilles Charvin (iD) https://orcid.org/0000-0002-6852-6952

## Decision letter and Author response
Decision letter https://doi.org/10.7554/eLife.34025.024
Author response https://doi.org/10.7554/eLife.34025.025

## Additional files

### Supplementary files
• Supplementary file 1. List of strains used in the study.
DOI: https://doi.org/10.7554/eLife.34025.019

• Supplementary file 2. Mean duration of cell phases in specific mutant backgrounds.
DOI: https://doi.org/10.7554/eLife.34025.020

• Transparent reporting form
DOI: https://doi.org/10.7554/eLife.34025.021

### Data availability
The raw cell cycle data are available on a dedicated server http://charvin.igbmc.science/yeastcycle-dynamics/ and further details on how to use the site are available in Appendix 1. The autotrack software is available at GitHub: https://github.com/gcharvin/autotrack

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

## Appendix 1

DOI: https://doi.org/10.7554/eLife.34025.022

# Autotrack: software for automated extraction of cell cycle phases

Autotrack is software developed during this study to automate the processing of time-lapse datasets. It is based on the project architecture of PhyloCell, which is frontend software previously developed by our group to process time-lapse images. Both programs are available for download at github: https://github.com/gcharvin.

The software allows parallel segmentation, tracking, and quantification of individual cells in yeast microcolonies. Information on both the cell and nucleus is used to ensure a reliable determination of cellular parentage (mother/bud relationships), which is mandatory for assessing cell growth during a complete cell cycle (see *Figure 1—figure supplement 6* for an overview of the pipeline). Finally, a specific routine is used to extract the duration of individual phases for each cell cycle, and a quality control routine is run to discard outliers. The steps are described in detail below.

## Segmentation and tracking

Segmentation of cellular contours is performed on phase contrast images using a modified Watershed algorithm, as previously described (1). Cells are then tracked over successive frames using an assignment-cost procedure solved with the Hungarian method. Segmentation of nuclei is performed by thresholding after background subtraction using morphological operators. Nuclei tracking is achieved as described for cellular contours.

## Parentage analysis

Although mother/bud parentage can *a priori* be established in the absence of additional markers, it is not exempt from uncertainties, leading to lineage errors. In contrast, using an additional nuclear marker greatly alleviates concerns about potential mother/bud assignments.

In practice, cellular and nuclear contours were combined so that nuclei were identified for each frame of interest. In the rare case when a nucleus overlapped with two cellular contours (e.g., nuclear lengthening during anaphase), a link between a cell and its bud was established. Otherwise, each time a nucleus appeared (following nuclear division), an optimization algorithm allowed us to identify the best neighbor nucleus as the 'mother' nucleus from which it derived (based on both spatial localization and timing of divisions). Therefore, this analysis allowed us to link newly appearing daughter cells to their mother.

By establishing the mother/daughter links, we could build a complete pedigree analysis and determine cell division timing by detecting the sudden drop in histone level upon anaphase and using it as a reference point.

## Cell cycle phase extraction procedure

Extraction of cell cycle phases was performed assuming a simple evolution of the pattern of histone level during a cell cycle. Following a sudden drop in total histone level corresponding to nuclear division, we expected the histone level to be constant during G1, then linearly increase during S phase, and plateau during the subsequent G2/M phase until the onset of anaphase. Therefore, we fitted the total histone fluorescence data to a piecewise linear fit (based on the BSFK optimization, least-square fitting with Free-Knot B-spline), which allowed us to retrieve the duration of each interval.

Determination of cytokinesis timing, which is impossible using a histone marker alone, was assessed independently with a septin protein (Cdc10-mCherry, see *Figure 1—figure supplement 5*) and was found to be very tightly correlated with anaphase. Therefore, for the

sake of simplicity, we assumed that the duration of the anaphase–cytokinesis interval was fixed (see main text), ignoring potential cell-to-cell variations that may, in turn, affect measurement of the duration of the subsequent G1 phase.

Based on this analysis, we extracted a list of variables for each cell cycle, as detailed below:

| Timing | Definition |
|---|---|
| $T_{div}$ | Cell cycle division time (min) |
| $T_{G1}$ | G1 phase duration (min) |
| $T_S$ | S phase duration (min) |
| $T_{G2/M}$ | G2/M phase duration (min) |
| $T_{Ana/Cyt}$ | Anaphase to cytokinesis duration (min) |
| Cell size | |
| $V_{birth}$ | Cell volume at birth (daughter) or division (mother) (fL) |
| $V_{G1}$ | Cell volume at the end of G1 phase (fL) |
| $V_S$ | Cell volume at the end of S phase (fL), excluding bud |
| $V_{G2/M}$ | Cell volume at the end of G2/M phase (fL), excluding bud |
| $V_{Ana/Cyt}$ | Cell volume at the end of cytokinesis phase (fL), excluding bud |
| $V_{bS}$ | Bud volume at the end of S phase (fL) |
| $V_{bG2/M}$ | Bud volume at the end of G2/M phase (fL) |
| $V_{bAna/Cyt}$ | Bud volume at the end of cytokinesis (fL) |
| Other | |
| MD | Boolean specifying whether cell is a mother or a daughter |
| Division | Replicative age of the cells (Daughters: 0; Mothers: 1, 2, 3, etc.) |
| $\mu_{unb.}$ | Linear growth rate during the unbudded period of the cell cycle |
| $\mu_{bud.}$ | Linear growth rate during the budded period of the cell cycle |
| Asy | Daughter/Mother volume ratio at division |

## Cell cycle phase verification procedure

We used criteria to exclude outliers due to errors in segmentation, tracking, assignment of buds to mother cells, and issues related to the fitting of histone level curves. In practice, we defined time intervals to keep cell cycle durations within an acceptable range, as shown in the table below. In addition, following a careful inspection of individual data points, we arbitrarily chose to discard cell cycles in which a small size at birth was concomitant with a short duration of G1 ($V_{birth} \times T_{G1}$). We also noticed a common error related to the detection of the linear ramp in histone level during S phase. To remove this error, we discarded cell cycles in which the durations of both G1 and G2/M were very short ($T_{G1} \times T_{G2/M}$). Cells without buds by the end of G2/M and Anaphase (VbG2 <0 and VbAna <0, respectively) were also discarded, as well as cells in which the daughter was more than twice the size of its mother at birth (Asy >2, presumably due to a wrong mother/bud association). Last, we removed all cell cycles in which the goodness of fit to the model was below a certain threshold, as defined by the chi-square ($\chi2 > 0.01$). In total, up to 63% of all cell cycles were retained for cells in the wild-type background (26,945 cell cycles, including 10,091 outliers), and similar values were obtained with mutants.

To evaluate the impact of this data filtering procedure on the measurements performed throughout this study, first, we have quantified the cell cycle rejection rate for each selection criterium (e.g. $\chi^2$, tdiv, tg1, etc.) mentioned above (see **Figure 1—figure supplement 7A**). For instance, this analysis showed that abnormal histone curve fitting contributes to 15% of the total rejected cell cycles (using the arbitrary threshold $\chi^2 = 0.01$, see **Figure 1—figure supplement 7A and B**). Then, we have compared the cell cycle timings of selected cell cycles versus the whole set of data (see **Figure 1—figure supplement 7C**). This analysis revealed

that data filtering mostly changes the duration of G1, by discarding cell cycles with abnormally short G1 in daughter cells. Therefore, this procedure is useful to get rid of major artefacts associated with the automated tracking and fitting procedure. However, we checked that data filtering does not qualitatively change the results related to compensatory growth, even though the values of the corresponding slopes are slightly different in the 'selected' versus 'all' cell cycle datasets (see *Figure 1—figure supplement 7D*).

| Variable | Acceptable range |
|---|---|
| $T_{div}$ | [45–300 min] |
| $T_{G1}$ | [0–300 min] |
| $T_S$ | [0–300 min] |
| $T_{G2/M}$ | [0–300 min] |
| $T_{Ana/Cyt}$ | [2–30 min] |
| $\mu_{unb.}$ | >0 (in daughter cells only) |
| $Vb_{G2/M}$ | >0 |
| $Vb_{Ana}$ | >0 |
| Asy | <2 |
| Other constraints | |
| $V_{birth} \times T_{G1}$ | >300 (fL x min) |
| $T_{G1} \times T_{G2/M}$ | >9 (min$^2$) |
| $\chi^2$ | <0.01 |

## Yeast Cycle Dynamics: a web-based application to explore mutants cell cycle data

Yeast Cycle Dynamics (YCD) is a tool accessible online (http://charvin.igbmc.science/yeastcycledynamics/index.php) to compare and analyze the data acquired during this study for both mutant and wild type strains.

Different mutants can be compared thanks to our dynamic graph system. This tool uses R to compute several analyses involving a selection of mutants and features. Mutant lines are selected using a dedicated interface that allows to display a short movie of the growing cells with a representation of their main characteristics. This makes it possible to visually estimate and compare the consequences of a mutation on the growing yeasts and their cell cycle. The analysis tool then allows to study the distribution of a mixture of variables for both mother and daughter cells on every selected lines. The results can be visualize as a distribution or a whiskers box plot.

This tool offers the possibility to study the correlation existing among these variables and see if they are conserved in the selected mutants. Here again, two graphs, a histogram and a correlogram, are available to depict these features.

The distance segregating mutants can be computed for all or every combination of variables. This way, it is possible to see which lines behave similarly for a given set of parameters. This tool uses the Mahalanobis distance which minimize the influence of the noisiest datasets for this measure.

Also, a principal component analysis module has been developed to better identify the variables that are the most discriminant among the mutants. Variables are represented as vectors which direction and length show their relative influence in the observed behaviors.

Last, it is possible to select specific yeasts from the database based on the volume of their body, bud and nucleus as well as the time duration for every phase of the cell cycle. The characteristics of the resulting cells can be further explored using their individual file. Each page shows the cell identity, its genotype, the number of reported divisions and a summary of all the variables recorded during the procedure. The system also generates an interactive graph showing the evolution of the fluorescence signal as well as the volume of the cell and

its bud over time. When yeasts have been tracked during several divisions, the user can display these features for every step to see how they evolve in time.

YCD is developed in PHP v.5 and JavaScript. The graphs are generated using R v.3 and the Rgraph library. Data are stored and accessed from a PostgreSQL v.9 database. The gene descriptions used in the system have been downloaded from the Saccharomyces Genome Database.

The whole dataset, including single cell data for each mutant, is available upon request.

## Model linking size compensation mechanisms to overall size homeostasis

### Model assumptions

The linear map model was used to characterize the dynamics of cell size over successive divisions (2). According to this framework, we let $V_n$ be the volume of a cell at division $n$ and we assume that the cell grows according to an affine law during a cell cycle:

$$F(V_n) = a\,V_n + b \tag{1}$$

with $a$ and $b$ being constant parameters that characterize the growth process. Therefore, if the cell divides symmetrically, the volume Vn +1 of the two sibling cells at division is given by:

$$V_{n+1} = \frac{1}{2}\,(a\,V_n + b) \tag{2}$$

In the case of budding yeast, division is asymmetrical; hence, assuming that Vn is the volume of the daughter cell, **Equation 2** becomes:

$$V_{n+1} = r\,(a\,V_n + b) \tag{3}$$

where $r$ represents the asymmetry factor ($0 < r < ½$).

The fixed point $V_{eq}$ (i.e., the steady state) of this linear map is given by:

$$V_{eq} = r\,(a\,V_{eq} + b) \tag{4}$$

which yields:

$$V_{eq} = \frac{r\,b}{1 - a\,r} \tag{5}$$

This indicates that a stable steady state is reached only when $a\,r < 1$. Under this condition, one can re-write **Equation 3** using **Equation 5** as:

$$V_{n+1} = p\,V_n + (1 - p)\,V_{eq} \tag{6}$$

where p=r $a$ characterizes the strength of convergence of the linear map. Indeed, $V_{n+1}$ appears as the weighted average between $V_n$ and the steady state value $V_{eq}$. Therefore, the lower the value of p ($0 < p < 1$), the more rapid the return to equilibrium (see **Figure 4**).

### Link to analyses based on ΔV vs. V plots

Most previous experimental analyses of size control have been based on measuring the variation of volume during the cell cycle as a function of the initial volume (ΔV vs. V plots) rather than on direct return map analyses.

Using the notations defined above, the variation of volume during a cell cycle can be written as:

$$\Delta V = a\, V_n + b - V_n = (a-1)\, V_n + \left(\frac{1}{r} - a\right) V_{eq} \tag{7}$$

thereby showing that the variation in cell volume varies linearly with the initial cell volume, with a slope referred to as $s_{tot}$ in the main text, such that: $s_{tot} = a - 1$

Interestingly, the measurement of size control based on the slope of $\Delta V$ vs. $V$ plots does not take the asymmetry factor into account, unlike the linear map model. For instance, a slope $s_{tot} = +1$, which is usually assumed to represent a perfect 'Timer' model (3), can still lead to size homeostasis provided that the asymmetry factor $r$ is smaller than ½, as in the case of budding yeast. Indeed, with $a = 2$ and $r < 1/2$, we get $p<1$, which ensures the convergence of the linear map, even though $s_{tot} = +1$.

## The noisy linear map

A Langevin noise term $\eta$, such that $<\eta>=0$ and $<\eta 2>=$constant, can be added to *equation (6)* to reflect the stochastic nature of the growth process and to investigate how this noise affects the distribution of cell size. In this context, *Equation 7* becomes:

$$V_{n+1} = p\, V_n + (1-p)\, V_{eq} + \eta \tag{8}$$

Whereas the mean cell volume is not changed in the presence of noise, noise generates fluctuations around the mean, the variance of which can be calculated using *Equation 8*:

$$
\begin{aligned}
<V^2> \; &\equiv <V_{n+1}^2> \\
&= <\left(p\,V_n + (1-p)V_{eq} + \eta\right)^2> \\
&= p^2<V^2> + p(1-p)V_{eq}^2 + (1-p)V_{eq}^2 + <\eta^2> \\
&= p^2<V^2> + (1+p)(1-p)V_{eq}^2 + <\eta^2>
\end{aligned}
$$

assuming $<V> = V_{eq}$. Therefore:

$$Variance\,(V) \equiv \; <V^2> - <V>^2 = \frac{<\eta^2>}{(1-p)(1+p)} \tag{9}$$

The Fano factor, which is a size-independent measure of fluctuations, is defined as:

$$Fano \equiv \frac{<V^2> - <V>^2}{<V>} = \frac{<\eta^2>}{V_{eq}} \frac{1}{(1-p)(1+p)} \tag{10}$$

Therefore, the Fano factor is the product of two terms: $<\eta^2>/V_{eq}$ characterizes the intrinsic noise in the growth process, whereas the other term is directly dependent on the strength of size control mechanism(s).

## Bayesian statistics analysis of links between cell cycle variables

A key result of this work is to show that there is a negative correlation between bud size at the end of $S$ phase ($Vb_S$) and G2/M duration ($T_{G2/M}$) which suggest the existence of a G2/M size control mechanism. However, the data also show that the duration of $S$ phase ($T_S$) correlates both with $Vb_S$ and $T_{G2/M}$ which could be untimely the underlying cause of an indirect correlation between $Vb_S$ and $T_{G2/M}$, hence the proposed underlying size control mechanism would be artefactual. These two scenarios can be depicted with the two simple tree belief networks shown below:

$$T_s \rightarrow V_{b_S} \rightarrow T_{G2/M} \tag{11}$$

and:

$$V_{b_S} \leftarrow T_S \rightarrow T_{G2/M} \tag{12}$$

which link conditional dependencies between variables. In **Equation 11**, the dependency between the variables $T_{G2/M}$ and $Vb_S$, and $Vb_S$ and $T_S$ are direct which induces an indirect correlation between the $T_{G2/M}$ and $T_S$. In contrast, the dependency between $T_{G2/M}$ and $T_S$ is direct in **Equation 12**.

The question is then whether it is possible to elucidate which of these two alternative models is correct. A first clue can already be obtained by having a closer look at the correlation coefficients between the three variables: $\rho(T_S, Vb_S) = 0.33$, $\rho(T_S, T_{G2/M}) = -0.29$ and $\rho(Vb_S, T_{G2/M}) = -0.43$ (after removing cell cycles in which Vb$_S$=0). The fact that the correlation between $Vb_S$ and $T_{G2/M}$ is stronger than the correlation between $T_S$ and $T_{G2/M}$ suggest that the conditional dependency $T_S \rightarrow Vb_S \rightarrow T_{G2/M}$ agrees better with the data.

To further support this statement, we use a Bayesian statistical approach that allows us to evaluate which tree belief network (**Equation 11** versus **Equation 12**) fits better the data(4). To do so we first discretize the three-dimensional space formed by the three cell-cycle variables considered here ($T_{G2/M}, Vb_S, T_S$) into voxels by splitting their domains into $b$ bins. We then and count the number of cells $n_{xyz}$ that lay inside each voxel labeled by the three discrete indexes $x, y, z$ (where the identities $x = T_{G2/M}$, $y = Vb_S$ and $z = T_S$ were introduced to simplify the notation). We then assume that there is a join discrete probability distribution $p_{xyz}$ that stands for how likely is to find a cell in the bin $x, y, z$. The two belief network models represent a specific factorization of the join distribution as a product of conditional probabilities. that is $p_{xyz} = p_{x|y}p_{y|z}p_z$ or $p_{xyz} = p_{x|z}p_{y|z}p_z$. Using a Dirichlet distribution with concentration parameters (or pseudocounts) $\alpha = (\alpha, \ldots, \alpha)$ as a prior, we can integrate out all the discrete distributions. The likelihood of the data $D \equiv \{n_{xyz}\}$ given the first belief network is:

$$
\begin{aligned}
P(D|T_S \rightarrow Vb_S \rightarrow T_{G2/M}) &= \int \prod_{xyz} \Gamma\left(p_{x|y}p_{y|z}p_z\right)^{n_{xyz}+\alpha-1} dp_{x|y}dp_{y|z}dp_z \\
&= \frac{\prod_{xy}\Gamma(n_{xy}+b\alpha)\prod_{yz}\Gamma(n_{yz}+b\alpha)}{\prod_y\Gamma(n_y+b^2\alpha)\Gamma(n+b^3\alpha)}.
\end{aligned}
$$

where the sums over indexes are indicated by removing them, for example: $n_{xy} \equiv \sum_z n_{xyz}$. Alternatively, given the second belief network we obtain:

$$
\begin{aligned}
P(D|Vb_S \leftarrow T_S \rightarrow T_{G2/M}) &= \int \prod_{xyz} \Gamma\left(p_{x|z}p_{y|z}p_z\right)^{n_{xyz}+\alpha-1} dp_{x|z}dp_{y|z}dp_z \\
&= \frac{\prod_{xz}\Gamma(n_{xz}+b\alpha)\prod_{yz}\Gamma(n_{yz}+b\alpha)}{\prod_z\Gamma(n_z+b^2\alpha)\Gamma(n+b^3\alpha)}.
\end{aligned}
$$

And finally, the ratio of the two probabilities leads to:

$$
r = \frac{P(D|T_S \rightarrow Vb_S \rightarrow T_{G2/M})}{P(D|Vb_S \leftarrow T_S \rightarrow T_{G2/M})} = \frac{\prod_{yz}\Gamma(n_{yz}+b\alpha)\prod_z\Gamma(n_z+b\alpha)}{\prod_{xz}\Gamma(n_{xz}+b^2\alpha)\prod_y\Gamma(n_y+b^2\alpha)}
$$

Notice that the ratio depends only on the counts $n_{yz}$ and $n_{xz}$ which are the ones related to the links of the networks that are different between the two models. that is $Vb_S \rightarrow T_{G2/M}$ and $T_S \rightarrow T_{G2/M}$. Then, applying the Bayesian theorem, we obtain that the probability of model $T_S \rightarrow Vb_S \rightarrow T_{G2/M}$ given the data is:

$$
\begin{aligned}
P(T_S \rightarrow Vb_S \rightarrow T_{G2/M}|D) &= \frac{P(D|T_S \rightarrow Vb_S \rightarrow T_{G2/M})}{P(D|T_S \rightarrow Vb_S \rightarrow T_{G2/M}) + P(D|Vb_S \leftarrow T_S \rightarrow T_{G2/M})} \\
&= \frac{r}{r+1}
\end{aligned}
$$

where we assume that the prior probabilities of each model are equal, that is $P(T_S \to Vb_S \to T_{G2/M}) = P(Vb_S \leftarrow T_S \to T_{G2/M}) = 1/2$.

Finally, substituting the data in the previous equation, that is the discrete counts $\{n_{xyz}\}$ that indicate the number of cells that showed values for the variables $T_{G2/M}$, $Vb_S$ and $T_S$ within the voxel $xyz$, we obtain $P(T_S \to Vb_S \to T_{G2/M}|D) = 1$ for different number of bins (10, 50 and 100) and pseudocounts ($10^{-6}$, 0.5 and 1). Namely, the conditional dependency structure $T_S \to Vb_S \to T_{G2/M}$ is consistently supported by the data indicating that the correlation between S and G2/M durations is indirect and the negative correlation between bud size at the end of S phase and G2/M duration is direct. It is important to note that this approach allows us to evaluate only conditional dependencies and not causation. Indeed, it is not possible to distinguish the models (**Equation 11** and **Equation 12**) from similar ones where the direction of the arrows are flipped. However, the fact that there is a temporal ordering of our variables (i.e. the size of the bud after S phase is determined before the duration of the G2/M phase is realized) allows to state that the link $Vb_S \to T_{G2/M}$ must be causative.

