## [Decision Letter]

Thank you for submitting your article "Multiple inputs ensure yeast cell size homeostasis during cell cycle progression" for consideration by *eLife*. Your article has been reviewed by Andrea Musacchio as the Senior Editor, a Reviewing Editor, and three reviewers. The reviewers have opted to remain anonymous.

The reviewers have discussed the reviews with one another and the Reviewing Editor has drafted this decision to help you prepare a revised submission.

Summary:

Garmendia-Torres et al., describe an elegant new approach to visualizing cell-cycle progression in budding yeast. Their work describes a brute-force experimental approach that allows for a thorough quantification of different cell cycle stages with respect to timing and cell volume. Using this method the authors investigated the effects of activators and repressors of cell cycle progression. The work provides strong indication that in yeast cell size homeostasis does not originate solely from a G1-specific mechanism, also from cell size regulation in later phases of the cell cycle, i.e. G2/M, and that it is likely to result from the integration of several mechanisms that coordinate cell growth with division, as well as asymmetric division. Specifically, the work highlights the role of bud size control in limiting cell-to-cell variability. The experimental method itself was thoroughly validated, providing high quality data for the subsequent analysis and the finding of a size control step in G2/M is highly interesting. Overall the manuscript is clearly written and the large single-cell dataset in particular, which is made available to the wider community via a dedicated web server, is a very valuable contribution. The work will be of great interest to a wide audience working in cell cycle, cell biology and related fields. Nonetheless, attention to the following issues would improve the manuscript.

Essential revisions:

1) The authors use a constant value derived from the septin-mCherry experiment to delineate anaphase from cytokinesis (and with this the begin of G1). To what extent this may affect the exploration of G1 phase related measurements is not clear, especially considering that some of the mutants tested may also affect cytokinesis. Some more comments here might be helpful.

2) The authors used a previously employed methodology where the change of one parameter is investigated as a function of another parameter, which, in the particular case, sensitizes the analysis towards particular behaviours, here the Timer, Sizer and Adder type of growth size control. This analysis discovered rather unexpectedly a relatively large size-compensation mechanism operating in G2/M, which is of comparable magnitude to the known Sizer during G1. I think this is an important point of the paper, which is relatively easy to miss, as it is convoluted with a lot of additional observations. The authors could consider to introduce more subheadings and to split Figure 3 to account for this.

3) Size compensation and subsequently noise measurements were conducted to investigate a series of mutants that all are linked somehow to cell cycle and cell size control. While this analysis seems to be rigorous, it is hard to comprehend if someone is not an absolute specialist in cell cycle regulation. Here, the paper could probably profit from a reduction of data – instead of all mutants only few and selected mutants with a more detailed discussion of the molecular mechanisms behind could help to guide the reader through the rather complex statistical analyses.

4) The authors employ a Htb2-fluorescent protein fusion as THE central tool of the entire work. GFP fused to other Histone variants have in the past also been used to discriminate cell cycle stages, e.g. Hhf2 in Newman et al., 2006. Histones, however, are essential proteins involved in chromatin functions. Most of them are expressed from redundant gene pairs, which makes it possible that the influence of the tag on the individual proteins is more severe than is expected from the slight growth delay observed. This could impact e.g. global gene regulation and, with this, cause pleiotropic phenotypes that could also manifest in problems associated with cell cycle, size and shape control. For example, on an empirical note (based on the reviewer's own experience), the Hhf2-GFP cells used in Newman et al., look quite 'abnormal'. Maybe the authors have tried out different histone variants? It would be nice if they could comment on this. Did the authors compare cell shape and size distributions etc. of their tagged variant with the wild type?

5) Some of the phrasings in the paper is convoluted and makes it quite difficult to rationalize. As an example, the sentence: “To determine how size G1 and G2/M compensation effects actually impact size homeostasis, we sought to directly assess the evolution of the variability in cell size during cell cycle progression." could be rephrased to: “To determine how size compensation effects in G1 and G2/M actually impact size homeostasis, we quantified cell size variability during cell cycle progression."

6) Figure 3D: Point one and also two (small buds) could be convoluted with a large measurement error. I am a bit concerned that the analysis here is convoluted by the precision by which the individual cell cycle stages for the measurements can be determined, and because with number the measured area becomes bigger and the relative size increase is small, which makes the measurements intrinsically more accurate. Some thoughts on this would be useful. Also, the color code used in this figure and incomplete axis labeling, along with rather cryptic description in the text make this figure a true challenge.

7) The fact that the authors filter out a third of their cells needs to be addressed (in the Supplemental Information) with some explanation of why they are confident that this loss of data does not bias their results.

8) In Figure 3C, it is unclear how s_tot_ can be independent of s_G1_ and s_G2/M_. In particular, how can s_G1_ and s_G2_ for swe1 cell both be about 0, but s_tot_ be 0.4?

9) The model suggests that *swe1 whi5* double mutant cells might be dead, since they should be defective in size compensation at both G1/S and M/A. Testing this suggestion is not an essential result for publication, but, if the phenotype of *swe1 whi5* cells has already been published, it should be mentioned. If not, it would be easy to determine and would add significantly to the paper.

10) The authors' conclusions that there is more than one point of size control in budding yeast is convincing, but they overstate their case by claiming that size control is "likely to be an emergent property resulting from the integration of several mechanisms". This claim strongly suggests that there are more than just the two size control targets (Start and mitosis) that they identify. As they point out, two redundant points of size control are well established in fission yeast. They need to explicitly state that such a situation would efficiently account for their data before speculating on more complicated possibilities.

11) The application of metazoan cell-cycle phase names (G1, S, G2, M) to budding yeast has confused the field for decades. In particular, the fact that budding yeast assemble a mitotic spindle (diagnostic of mitosis in other organisms) before they complete replication mean that S phase and M phase overlap, making G2 (the time between the end of S and the beginning of M in other organisms) mean something different in budding yeast. The authors are not going to fix this problem, but it would be a great help to their readers if they mitigated it in the following two instances.

a) First, they should point out that G2/M is equivalent to metaphase (the period after spindle formation and before anaphase) in other organisms.

b) Second, they should be careful with the term "Mitotic Entry". They use it to refer to the G2/M transition in fission yeast, which is appropriate, but, in the section header“Control of Mitotic Entry via a Bud-Specific Size Compensatory Mechanism”, they appear to use it to refer to the M/A transition in budding yeast, which is problematic because this transition is widely (if confusingly) referred to a mitotic exit in the budding yeast literature. Elsewhere, they describe Swe1 as an inhibitor of mitotic entry, but it inhibits CDK activation which regulates the G2/M transition in other organisms (and presumably spindle formation in budding yeast, although this regulation has not been well characterized) not the M/A transition. I think the authors would be best served not using the term mitotic entry to refer to budding yeast, but rather explicitly refer to the particular transitions, i.e. M/A.

12) Insights about this mechanism are somewhat buried under the statistical analysis in the last part of the paper, while the biochemical mechanism that establishes G2/M size control is only superficially addressed. In particular, while the mutant strains point to a role of Cyclin B-Cdk activation in G2/M size control, how exactly this control is implemented and which of the various molecules that influence Cyclin B-Cdk activation are involved remains elusive. The authors could have utilised their method to study the mechanistic basis of G2/M size control in more detail. Specifically, an important concern is related to the conclusion that cell size also affects G2/M progression, with G2/M control not being cryptic, i.e., it is observed in wild-type cells under normal growth condition, and its strength being equal if not stronger than G1 control. In Figure 1D, the authors present the duration of various cell cycle phases and their distribution. These data clearly show a difference between large, mother cells (red) and small, daughter cells (blue) for G1 as well as S-phase, albeit with a lower magnitude. No substantial difference was observed for the duration of G2/M. Similarly, the variation (indicated by the standard error on mean; note the wrong legend for the leftmost plot of Figure 1D) is highest for G1 phase in daughter cells followed by the daughters' S phase, with G2/M phase showing a low variability. Together these data clearly point towards G1 (and not G2/M) size control. In addition, G2/M phase is much shorter than G1 phase, which makes up more than half of the total cell cycle time, potentially allowing for a much larger adjustment to size differences. It is unclear how these observations fit to the main conclusion of an important G2/M control step, but the suspicion is that it might be cryptic in this case. Perhaps the author could provide some discussion on this and a figure similar to 1D for mutants in which G1 and G2/M size control are particularly strong or weak.

---

## [Author Response]

Summary:Garmendia-Torres et al., describe an elegant new approach to visualizing cell-cycle progression in budding yeast. Their work describes a brute-force experimental approach that allows for a thorough quantification of different cell cycle stages with respect to timing and cell volume. Using this method the authors investigated the effects of activators and repressors of cell cycle progression. The work provides strong indication that in yeast cell size homeostasis does not originate solely from a G1-specific mechanism, also from cell size regulation in later phases of the cell cycle, i.e. G2/M, and that it is likely to result from the integration of several mechanisms that coordinate cell growth with division, as well as asymmetric division. Specifically, the work highlights the role of bud size control in limiting cell-to-cell variability. The experimental method itself was thoroughly validated, providing high quality data for the subsequent analysis and the finding of a size control step in G2/M is highly interesting. Overall the manuscript is clearly written and the large single-cell dataset in particular, which is made available to the wider community via a dedicated web server, is a very valuable contribution. The work will be of great interest to a wide audience working in cell cycle, cell biology and related fields. Nonetheless, attention to the following issues would improve the manuscript.Essential revisions:1) The authors use a constant value derived from the septin-mCherry experiment to delineate anaphase from cytokinesis (and with this the begin of G1). To what extent this may affect the exploration of G1 phase related measurements is not clear, especially considering that some of the mutants tested may also affect cytokinesis. Some more comments here might be helpful.

Thanks for raising this point. The hypothesis suggested here is that some mutants may have a delayed cytokinesis that may artificially lengthen the G1 interval in our study. We cannot rule out this possibility. Yet, to the best of our knowledge, the vast majority of the mutants tested here has not been reported to alter cytokinesis timings.

There is, however, one notable exception with the *cdh1****Δ*** mutant, which displays defects in actomyosin disassembly (Tully et al., 2008). In this case, a delay in the progression through cytokinesis of ~10 minutes is observed. However, it is unclear how this delay should affect the duration of the next G1, because this interval is controlled by cell growth/size and growth may not be impaired in cells experiencing delays during cytokinesis. In practice, in agreement with the literature, we measured a reduced (i.e. not increased, as the hypothesis raised above would suggest) G1 duration in the *cdh1****Δ*** mutant, indicating that (1) Our methodology captured the previously reported G1 duration phenotype correctly for this mutant and (2) The G1 duration phenotype of this mutant is most likely explained by its inhibition of premature Cdk activity in G1 rather than its indirect effect on cytokinesis.

Therefore, overall, we think the methodological approximation made in our study is well justified for all mutants reported. However, since this is a fair comment, we have added a sentence in the manuscript to mention that our methodology might introduce artifacts regarding G1 duration measurement in some particular mutant backgrounds.

(Interestingly, unlike cytokinesis defects, we have shown that a *slk19*D mutant (Slk19 is a kinetochore-associated protein) displays a t_Ana_ interval longer than WT (17 minutes and 12 minutes respectively), due to defects in chromosome segregation, in agreement with the literature. This shows that perturbations of this particular cell cycle interval can also be captured using our methodology.)

2) The authors used a previously employed methodology where the change of one parameter is investigated as a function of another parameter, which, in the particular case, sensitizes the analysis towards particular behaviours, here the Timer, Sizer and Adder type of growth size control. This analysis discovered rather unexpectedly a relatively large size-compensation mechanism operating in G2/M, which is of comparable magnitude to the known Sizer during G1. I think this is an important point of the paper, which is relatively easy to miss, as it is convoluted with a lot of additional observations. The authors could consider to introduce more subheadings and to split Figure 3 to account for this.

We thank the reviewer for this suggestion. To emphasize our findings related to G2/M bud size control, we have split Figure 3 in three parts (new Figure 3, new Figure 4, and new Figure 5) that corresponds to the following subsections respectively: “Control of the metaphase to anaphase transition via a Bud-Specific Size Compensatory Mechanism”, “Impaired Size Control in Mutants of Cyclin B Regulation and Function”, and “Effective Cell Size Homeostasis during Cell Cycle Progression”. Therefore, in the revised version, the new Figure 3 only illustrates the conclusion of the first section, which explicitly comments on the magnitude of the G2/M size control. We believe that these changes improve the readability of the manuscript and emphasizes this important finding.

3) Size compensation and subsequently noise measurements were conducted to investigate a series of mutants that all are linked somehow to cell cycle and cell size control. While this analysis seems to be rigorous, it is hard to comprehend if someone is not an absolute specialist in cell cycle regulation. Here, the paper could probably profit from a reduction of data – instead of all mutants only few and selected mutants with a more detailed discussion of the molecular mechanisms behind could help to guide the reader through the rather complex statistical analyses.

We acknowledge that the large number of mutants reported in our study could be confusing to non-specialist readers. However, we would like to draw the attention of the reviewer to the fact that Figure 2 and Figure 3 already display the results obtained with a subset of mutants (12 in Figure 2, 15 in new Figure 4, out a of total of 22 mutants). Data obtained with other mutants are available on the supporting online database.

In addition, we think that the large number of mutants displayed in new Figure 4 (based on a selection of key cell cycle regulators) is essential to assess the magnitude of correlation between Fano factor fold-change and magnitude of the compensatory growth.

Last, regarding the last comment of the reviewer, even though a longer discussion dedicated to the mechanism that may link a specific gene function to size compensation phenomena might be useful, we think that it goes beyond the scope of this study and would be too speculative. Indeed, in particular, we believe that our study will trigger complementary work attempting to decipher the underlying mechanism enabling the G2/M size control that was reported in the present manuscript.

4) The authors employ a Htb2-fluorescent protein fusion as THE central tool of the entire work. GFP fused to other Histone variants have in the past also been used to discriminate cell cycle stages, e.g. Hhf2 in Newman et al., 2006. Histones, however, are essential proteins involved in chromatin functions. Most of them are expressed from redundant gene pairs, which makes it possible that the influence of the tag on the individual proteins is more severe than is expected from the slight growth delay observed. This could impact e.g. global gene regulation and, with this, cause pleiotropic phenotypes that could also manifest in problems associated with cell cycle, size and shape control. For example, on an empirical note (based on the reviewer's own experience), the Hhf2-GFP cells used in Newman et al. look quite 'abnormal'. Maybe the authors have tried out different histone variants? It would be nice if they could comment on this. Did the authors compare cell shape and size distributions etc. of their tagged variant with the wild type?

In agreement with the reviewer’s experience, we have attempted to work with an Hhf2-GFP-tagged strain and we have observed significantly longer cell cycle duration (almost 2 hours in normal growth conditions) and larger cell size (more than twice as big as an Htb2-GFP strain), indicating that cell cycle progression in this strain is far from physiological conditions. In addition, the replicative lifespan is also shorter in this background than WT (14 generations and 25 generations, respectively). Last, we have obtained similar results with Hhf1.

Interestingly, most live imaging studies in the literature use H2B as a histone-GFP reporter. It is likely that tagging this particular histone induces less physiological alterations than with other variants.

5) Some of the phrasings in the paper is convoluted and makes it quite difficult to rationalize. As an example, the sentence: “To determine how size G1 and G2/M compensation effects actually impact size homeostasis, we sought to directly assess the evolution of the variability in cell size during cell cycle progression." could be rephrased to: “To determine how size compensation effects in G1 and G2/M actually impact size homeostasis, we quantified cell size variability during cell cycle progression."

We thank the reviewer for this specific suggestion and we have introduced this change in the revised version. More generally, we have made several changes in the main text to improve the readability.

6) Figure 3D: Point one and also two (small buds) could be convoluted with a large measurement error. I am a bit concerned that the analysis here is convoluted by the precision by which the individual cell cycle stages for the measurements can be determined, and because with number the measured area becomes bigger and the relative size increase is small, which makes the measurements intrinsically more accurate. Some thoughts on this would be useful. Also, the color code used in this figure and incomplete axis labeling, along with rather cryptic description in the text make this figure a true challenge.

We thank the reviewer for raising this point. Indeed, error on volume measurement was not estimated in the original submission, and we agree with the reviewer that this error is likely to be proportionally higher in small cells than in large ones.

Volume measurement was based on the automated segmentation of cellular contours using custom software. To get an estimate of the error associated with this procedure, we carefully performed a manual (blind) segmentation of more than 500 cells with various sizes, which we considered as ground truth. Then, for each manually segmented cell, we estimated the volume measurement error as the difference between the volume calculated from automated segmentation and that calculated from the ground truth – this actually provides an upper bound, knowing that manual segmentation necessarily introduces additional errors.

Then, we calculated the mean relative volume error (i.e. experimental error divided by mean cell volume) after pooling cells in groups of increasing size (new Figure 5A). We found that the mean relative volume error decreases with cell size, as expected by the reviewer, and we observed that the scaling is similar to that observed for the variability in cell volume (i.e. 1/sqrt(volume)). However, the amplitude of this experimental noise appeared much lower than that cell size noise (compare Fano factor for experimental error versus data). This shows that the measurement of cell volume noise, as performed in our study, truly reflects the actual cell-cell variability observed in a population of cells and is not dominated by measurement error. The result of this analysis is displayed on Figure 5A in the revised version and shows a direct comparison between intrinsic cell volume noise and experimental measurement noise.

Last, we have redesigned panel A and extensively revised the legend of this panel to take the reviewer’s comment into account regarding the readability of the figure.

7) The fact that the authors filter out a third of their cells needs to be addressed (in the Supplemental Information) with some explanation of why they are confident that this loss of data does not bias their results.

We agree with the reviewer that details related to data filtering and its impact on the robustness of the results were missing in the original submission.

In the revised version, we have updated the supplemental text and made a new supplemental figure (Figure 1—figure supplement 7) to address this point: first, we have quantified the data rejection rate for each selection criterium (e.g. chi2, tdiv, tg1, etc.) that is mentioned in the supplemental text (Figure 1—figure supplement 7A). This analysis allows one to understand why cell cycles are or are not selected (e.g. a threshold in chi2 is used to discard bad fits, see Figure 1—figure supplement 7B). Second, we have compared the cell cycle timings of selected cell cycles versus the whole dataset (Figure 1—figure supplement 7C). This analysis shows that data filtering mostly changes the mean duration of G1, by discarding cell cycles with abnormally short G1 in daughter cells. However, it does not qualitatively change the results related to compensatory growth, even though the values of the corresponding slopes are slightly different in the filtered versus whole dataset (Figure 1—figure supplement 7D).

Using this complementary analysis, we demonstrate that the data filtering procedure is useful to get rid of artefacts associated with histone level quantification and fitting, but it does not affect the key observations reported in our study.

8) In Figure 3C, it is unclear how s_tot_ can be independent of s_G1_ and s_G2/M_. In particular, how can s_G1_ and s_G2_ for swe1 cell both be about 0, but s_tot_ be 0.4?

We thank the reviewer for raising this point. Overall, Figure 3D shows that mutants with stronger G1 and/or G2/M compensatory growth roughly display a stronger overall compensatory growth over an entire cell cycle.

However, s_tot_ cannot be simply deduced by averaging/combining the slopes associated with G1 and G2/M measurements. For instance, WT mother slopes are -0.02 and -0.49 for G1 and G2/M, respectively, yet s_tot_ is slightly positive (+0.09, Figure 3D). It is possible that other phases of the cycle (e.g. S-phase) behave as Timers and thus contribute to increase s_tot_. Then, the overall compensatory growth is likely to depend on the respective duration of each phase. Last, unlike that of G1, the measurement G2/M compensatory growth is using the size of the bud as a reference of size, which makes the averaging of slope even less relevant.

In the specific case of the *swe1* mutant, switching to a robust regression procedure induced a slight decrease in s_tot_ (from 0.4 to 0.15), hence this slope turns out to be closer to s_G1_ (-0.12) and s_G2/M_ (-0.15), even though the sign is different.

9) The model suggests that swe1 whi5 double mutant cells might be dead, since they are should be defective in size compensation at both G1/S and M/A. Testing this suggestion is not an essential result for publication, but, if the phenotype of swe1 whi5 cells has already been published, it should be mentioned. If not, it would be easy to determine and would add significantly to the paper.

This is an interesting suggestion by the reviewer. Due to the large collection of mutants already reported in the present manuscript (see point #3 above), we have chosen not to perform experiments with double mutants. However, we agree that synergistic effects of mutations should be specifically assessed in a complementary study in order to get better mechanistic insights about the G2/M size control unraveled in the present manuscript. Last, to the best of our knowledge, the phenotype of *swe1 whi5* has not been reported.

10) The authors' conclusions that there is more than one point of size control in budding yeast is convincing, but they overstate their case by claiming that size control is "likely to be an emergent property resulting from the integration of several mechanisms". This claim strongly suggests that there are more than just the two size control targets (Start and mitosis) that they identify. As they point out, two redundant points of size control are well established in fission yeast. They need to explicitly state that such a situation would efficiently account for their data before speculating on more complicated possibilities.

As pointed by the reviewer (and mentioned in the introduction of our manuscript), the existence of two distinct points of size control has already been reported in fission yeast (in G1 and G2). However, in this case, the G1 size control has been characterized as a “cryptic” phenomenon that is manifest in specific mutant backgrounds, such as wee1, but is absent in WT. Unlike this paradigm in which one checkpoint is dominant and the other one behaves as a “backup” mechanism, our study reveals that two mechanisms of size control operate in different of phases of the cell cycle and both contribute to lower cell size variability in WT cells. We think that this distinction is a key aspect of our study. Therefore, we have added a couple of sentences in the discussion to emphasize the biological significance of our results. We have also changed “several” for “at least two” in the sentence highlighted by the reviewer.

11) The application of metazoan cell-cycle phase names (G1, S, G2, M) to budding yeast has confused the field for decades. In particular, the fact that budding yeast assemble a mitotic spindle (diagnostic of mitosis in other organisms) before they complete replication mean that S phase and M phase overlap, making G2 (the time between the end of S and the beginning of M in other organisms) mean something different in budding yeast. The authors are not going to fix this problem, but it would be a great help to their readers if they mitigated it in the following two instances.

We thank the reviewer for raising this issue. We have specifically addressed each of the following points in the revised version, as explained below.

a) First, they should point out that G2/M is equivalent to metaphase (the period after spindle formation and before anaphase) in other organisms.

We agree with the recommendation of the reviewer. The revised version has been modified to clarify the biological meaning of this particular time interval.

b) Second, they should be careful with the term "Mitotic Entry". They use it to refer to the G2/M transition in fission yeast, which is appropriate, but, in the section header “Control of Mitotic Entry via a Bud-Specific Size Compensatory Mechanism”, they appear to use it to refer to the M/A transition in budding yeast, which is problematic because this transition is widely (if confusingly) referred to a mitotic exit in the budding yeast literature. Elsewhere, they describe Swe1 as an inhibitor of mitotic entry, but it inhibits CDK activation which regulates the G2/M transition in other organisms (and presumably spindle formation in budding yeast, although this regulation has not been well characterized) not the M/A transition. I think the authors would be best served not using the term mitotic entry to refer to budding yeast, but rather explicitly refer to the particular transitions, i.e. M/A.

We acknowledge that our statements related to “mitotic entry” were inappropriate in this context. In the revised version, we have changed it for metaphase/anaphase transition in agreement with the reviewer’s suggestion.

12) Insights about this mechanism are somewhat buried under the statistical analysis in the last part of the paper, while the biochemical mechanism that establishes G2/M size control is only superficially addressed. In particular, while the mutant strains point to a role of Cyclin B-Cdk activation in G2/M size control, how exactly this control is implemented and which of the various molecules that influence Cyclin B-Cdk activation are involved remains elusive. The authors could have utilised their method to study the mechanistic basis of G2/M size control in more detail. Specifically, an important concern is related to the conclusion that cell size also affects G2/M progression, with G2/M control not being cryptic, i.e., it is observed in wild-type cells under normal growth condition, and its strength being equal if not stronger than G1 control. In Figure 1D, the authors present the duration of various cell cycle phases and their distribution. These data clearly show a difference between large, mother cells (red) and small, daughter cells (blue) for G1 as well as S-phase, albeit with a lower magnitude. No substantial difference was observed for the duration of G2/M.

We think that there may be a misunderstanding regarding the interpretation of data in Figure 1D and old Figure 3B (i.e. new Figure 3D), so this point is addressed in detail below to clarify.

The difference in G1 duration between mother and daughter cells (as reported in Figure 1D) is clearly due to the smaller size of daughters compared to mothers upon birth/division (see X axis of scatter plots showing V_birth_ for daughters and V_division_ for mothers on new Figure 3D). This smaller size is compensated by a larger volume increase during G1 (**Δ**V_G1_) in daughters compared to mothers (new Figure 3D).

Regarding the G2/M size control, we draw the attention of the reviewer to the fact that this control operates on the bud and not on the overall cell size at the end of S-phase (Figure 3—figure supplement 1). Figure 3D clearly shows that bud sizes by the end of S-phase are almost *identical* in mothers and daughters, and so the variations in bud size during G2/M (**Δ**V_G2/M_). Therefore, we expect G2/M durations to be *identical* in mothers versus daughters (and indeed they are, as pointed by the reviewer, and reported on Figure 1D). A critical feature of this G2/M size control is thus that it operates similarly in both mothers and daughters, unlike the G1 size control.

In summary, G1 size control has long been associated with observations of distinct G1 durations for mothers versus daughters, because these cells have different sizes upon birth/division. The fact the G2/M intervals are identical in mothers and daughter does mean that there is no size control in this interval, as indeed demonstrated in old Figure 3B (new Figure 3D) and explained above.

Similarly, the variation (indicated by the standard error on mean; note the wrong legend for the leftmost plot of Figure 1D) is highest for G1 phase in daughter cells followed by the daughters' S phase, with G2/M phase showing a low variability. Together these data clearly point towards G1 (and not G2/M) size control.

First, we apologize for the confusion due to the swapping of mothers/daughters legends on Figure 1D. This error has been fixed in the revised version.

Then, we think that the variability in cell cycle phase duration is not a reliable way to compare the magnitude of growth compensation during a particular cell cycle phase (a fortiori based on the standard error on mean). For instance, the coefficient of variation (CV) of G1 duration are 0.50 and 0.53 in mothers and daughters, respectively (measurements based on Figure 1D), yet it is clear that the magnitude of G1 size control is completely different in mothers versus daughters (new Figure 3D).

Rather, the consensus in the field of size control is to use the slope of the **Δ**V vs V plots as a readout (as reported in new Figure 3D), because it provides a direct measurement of compensatory growth during specific phases of the cell cycle, which is a signature of size control. Using this methodology, G2/M bud size control appears quite clearly (new Figure 3D), and the magnitude of compensatory growth is actually stronger to G1.

In addition, G2/M phase is much shorter than G1 phase, which makes up more than half of the total cell cycle time, potentially allowing for a much larger adjustment to size differences. It is unclear how these observations fit to the main conclusion of an important G2/M control step, but the suspicion is that it might be cryptic in this case. Perhaps the author could provide some discussion on this and a figure similar to 1D for mutants in which G1 and G2/M size control are particularly strong or weak.

We understand that the reviewer is concerned that the G2/M size control unraveled in our present study may have much less impact on the overall size homeostasis than the long described G1 size control. Yet, we believe our results provide solid arguments in favor of a strong physiological role of the G2/M size control, as explained below.

In WT, the slope of the **Δ**V vs V plot is stronger for G2/M compared to G1 (Figure 3B). Therefore, the strength of the corresponding growth compensation effect is larger or equal for buds in G2/M than for newborn cells in G1.

This reveals the existence of a size control mechanism operating in G2/M, but it does not prove that it contributes to control the overall size homeostasis during the cell cycle.

To demonstrate it, we measured the evolution of Fano factor during cell cycle progression and we show that it declines significantly during G2/M (and more than during G1, new Figure 5A-C). This is direct evidence that noise in cell size is reduced during G2/M. Last, we show that mutants in which G2/M growth compensation is greatly altered show lower noise reduction in G2 (Figure 5A-C) and display a larger overall size variability (Figure 6B).

We would like to draw the attention of the reviewer to the fact that most previous studies focused mainly on the magnitude of growth compensation effects yet overlooked the measurements of actual size homeostasis during the cell cycle, unlike our present work. Such assay provides a clear assessment of the functional incidence of a particular size control mechanism, as requested by the reviewer.

Last, regarding the reviewer’s last sentence, as explained above, we think that distributions (and the variability thereof) of phase durations are not appropriate readouts of the magnitude of size control in WT and mutants.